# Quantum field theory and the Bieberbach conjecture

Parthiv Haldar$^{\alpha *}$, Aninda Sinha$^{\alpha \dagger}$, and Ahmadullah Zahed$^{\alpha \ddagger}$

$^{\alpha}$*Centre for High Energy Physics, Indian Institute of Science,*
*C.V. Raman Avenue, Bangalore 560012, India.*

## Abstract

An intriguing correspondence between ingredients in geometric function theory related to the famous Bieberbach conjecture (de Branges' theorem) and the non-perturbative crossing symmetric representation of 2-2 scattering amplitudes of identical scalars is pointed out. Using the dispersion relation and unitarity, we are able to derive several inequalities, analogous to those which arise in the discussions of the Bieberbach conjecture. We derive new and strong bounds on the ratio of certain Wilson coefficients and demonstrate that these are obeyed in one-loop $\phi^4$ theory, tree level string theory as well as in the S-matrix bootstrap. Further, we find two sided bounds on the magnitude of the scattering amplitude, which are shown to be respected in all the contexts mentioned above. Translated to the usual Mandelstam variables, for large $|s|$, fixed $t$, the upper bound reads $|\mathcal{M}(s,t)| \lesssim |s^2|$. We discuss how Szegö's theorem corresponds to a check of univalence in an EFT expansion, while how the Grunsky inequalities translate into nontrivial, nonlinear inequalities on the Wilson coefficients.

# Contents

$^*$parthivh@iisc.ac.in
$^\dagger$asinha@iisc.ac.in
$^\ddagger$ahmadullah@iisc.ac.in

# 1 Introduction

In mathematics, the *Bieberbach conjecture* is about how fast the Taylor expansion coefficients of a holomorphic univalent[1] function, of a single complex variable $z$, on the unit disc ($|z| < 1$) grows. If we write this function as

$$f(z) = \sum_{n=1}^{\infty} b_n z^n \,, \tag{1.1}$$

then according to this conjecture

$$|b_n| \le n |b_1| \,. \tag{1.2}$$

This famous conjecture was put forth by Bieberbach in 1916 [1] and resisted a complete proof until 1985 when it was proved by de Branges [2]. Another important property for such univalent functions is what is known as the *Koebe Growth theorem* which says that

$$\frac{|b_1 z|}{(1 + |z|)^2} \le |f(z)| \le \frac{|b_1 z|}{(1 - |z|)^2} \,, \tag{1.3}$$

providing a two-sided bound on the absolute value of the function. In the course of 70 years, attempts at proving the Bieberbach conjecture led to the invention of new mathematical results such as (1.3) and techniques in the area of geometric function theory.

Now it is certainly not obvious, but we claim that (1.2) and (1.3) have analogues in the context of 2-2 scattering in quantum field theory. To see this, we will make use of the crossing symmetric representation of the 2-2 scattering of identical massive scalars, first used in the long-forgotten work by Auberson and Khuri [3] and resurrected in [4, 5]. If we consider $\mathcal{M}(s, t)$ and assume that there

---

[1]A function is univalent on a domain $D$ if it is holomorphic, and one-to-one, i.e. for all $z_1, z_2 \in D$, $f(z_1) \ne f(z_2)$ if $z_1 \ne z_2$.

are no massless exchanges, then we expect a low energy expansion of the form

$$\mathcal{M}(s,t) = \sum_{p\geq 0, q\geq 0} \mathcal{W}_{pq} x^p y^q \,, \tag{1.4}$$

where $x$ and $y$ are the quadratic and cubic crossing symmetric combinations of $s, t, u$ to be made precise below. Normally, the term dispersion relation for scattering amplitude $\mathcal{M}(s,t)$ refers to the fixed-$t$ dispersion relation, which lacks manifest crossing symmetry. As we will review below, in order to exhibit three-channel crossing symmetry in the dispersion relation, we should work with a different set of variables, $z$ and $a \equiv y/x$. For now, we note that both $x, y \propto z^3/(z^3 - 1)^2$. As such, the appropriate variable is $\tilde{z} = z^3$. We write a crossing symmetric dispersion relation in the variable $\tilde{z}$ keeping $a$ fixed. This dispersion relation, together with unitarity, leads to similar bounds as in (1.2) for the $\tilde{z}$ expansion of $\mathcal{M}(\tilde{z}, a)$ and as in (1.3) for $|\mathcal{M}(\tilde{z}, a)|$. The expansion of $M(\tilde{z}, a)$ around $\tilde{z} = 0$ is similar to a low energy expansion and the bound (1.2) relates to bounds on the Wilson coefficients $\mathcal{W}_{pq}$.

Over the last few months, the existence of upper and lower bounds on ratios of Wilson coefficients have been discovered [9, 10, 4]. These bounds are remarkable since they say that Wilson coefficients cannot be arbitrarily big or small and, in a sense, corroborate the efficacy of effective field theories. One of the interesting outcomes of our analysis is that

$$-\frac{9}{4\mu + 6\delta_0} < \frac{\mathcal{W}_{0,1}}{\mathcal{W}_{1,0}} < \frac{9}{2\mu + 3\delta_0} \,, \tag{1.5}$$

where $\mu = 4m^2$ with $m$ being the mass of the external scalar and $\delta_0$ is some cutoff scale in the theory. Expanding $\delta_0 \gg \mu$ leads to the 2-sided, space-time dimension independent bound $-\frac{3}{2\delta_0} < \frac{\mathcal{W}_{0,1}}{\mathcal{W}_{1,0}} < \frac{3}{\delta_0}$. Compared to [9, 10], the lower bound is identical but the upper bound we quote above is stronger. We have checked this inequality for several known examples. Other fascinating consequences of the analogs of (1.2) will be discussed below.

Univalence of a function leads to further nontrivial constraints in the form of the Grunsky inequalities, which are necessary and sufficient for an analytic function on the unit disk to be univalent. If these were to hold in QFT, they would imply non-linear constraints on $\mathcal{W}_{pq}$. Unfortunately, proving univalence is a tough problem. The $\tilde{z}$ dependence in the crossing symmetric dispersion relation arises entirely from the crossing symmetric kernel. One can show that this kernel, for a range of real $a$ values, is indeed univalent! Therefore, one concludes that for unitary theories, the amplitude is a convex sum of univalent functions. However, a complete classification of circumstances as to when a convex sum of univalent functions leads to a univalent function does not appear to be known in the mathematics literature. Nevertheless, just by using the univalence of the kernel, we will be able to derive analogues of (1.2) and (1.3). What we will further show is that as an expansion around $a \sim 0$, the Grunsky inequalities hold as the resulting inequality on $\mathcal{W}$ is known to hold using either fixed-$t$ or crossing symmetric dispersion relation. Therefore, at least around $a \sim 0$, it is indeed true that the amplitude, and not just the kernel in the crossing symmetric dispersion relation, is univalent. Our numerical checks for known S-matrices, such as 1-loop $\phi^4$, $\pi^0\pi^0 \to \pi^0\pi^0$ arising from the S-matrix bootstrap, tree-level string theory, suggest that there is always a finite region near $a \sim 0$ where univalence holds. Thus we conjecture that we can impose univalence on the amplitude even beyond the leading order in $a$. This gives rise to a non-linear inequality for the Wilson coefficients; as a sanity check, this inequality is satisfied for all the cases studied in this paper.

The discussion above may seem to suggest that we may need to know the full amplitude in QFT

to check for this seemingly magical property of univalence. Fortunately, this is not the case. In QFT, we would like to work in an effective field theory framework where we have access to certain derivative order in the low energy expansion. Thus we would need to know about the analogous statement in the mathematics literature, which deals with partial sums (truncations) of $f(z)$. Indeed there is such a theorem, called Szego's theorem! This remarkable theorem allows us to examine univalence for partial sums and, loosely speaking, states that the radius of the disk within which univalence holds for the partial sums of a function $f(z)$, which is univalent inside the unit disk, is at least $1/4$. We will use this theorem to rule out situations where univalence fails.

We should add that we are not the first to discuss univalence in physics: however, such discussion is scarce in the literature. To the best of our knowledge, in the context of high energy scattering amplitudes, such an investigation was first undertaken in the mid-1960s by Khuri and Kinoshita [6]. In more recent times, possible use of univalence of complex function has been discussed in the context of scattering amplitudes in [7] and in the context of bounding transport coefficients using AdS/CFT in [8]. We will review all three papers in an appendix. However, we want to emphasize that our treatment of univalence is quite different from all of these, as will become apparent in due course of time.

Let us now lay out the organization of the paper. First, we give a survey of the various aspects of univalent functions relevant to our analysis, in section 2. Next, in section 3, we review the crossing-symmetric dispersion relation and associated structures. Following this, we discuss the bounds on the Taylor coefficients of the scattering amplitude in section 4 the physical implications of which for Wilson coefficients is discussed in section 5. Next, we derive two-sided bounds on the scattering amplitude in section 6. In section 7, we look for hints of univalence in EFT amplitudes with the aid of Szegö's theorem followed by an exploration of Grunsky inequalities for amplitudes in section 7.1. Finally, we discuss the conclusions and provide outlooks on future directions in section 8. Various explorations associated with the main text providing the analysis with wholesomeness have been placed in the appendices.

# 2 Univalent functions and de Branges's theorem: A survey

A central theme of complex analysis is to study a complex function by the nature of the mapping produced by the function. A complex function $w = f(z)$ can be geometrically viewed as a mapping from a region in $z-$plane to $w-$plane, defined by $u = u(x, y)$ and $v = v(x, y)$, where $z = x + iy$ and $w = u + iv$. This aspect of complex analysis is known as "Geometric Function Theory". In geometric function theory, a class of functions called **univalent functions** play particularly important role. These functions will play a central role in our subsequent QFT analysis. Therefore, we will survey the crucial aspects of the univalent function in this section.

## 2.1 Univalent and schlicht functions

A function $f$ is defined to be *univalent* on a domain[2] $D \subset \mathbb{C}$ if it is holomorphic[3] and injective. The function is said to be *locally univalent* at a point $z_0 \in D$ if it is univalent in some neighbourhood of $z_0$. The function $f$ is locally univalent at some $z_0 \in D$ if and only if $f'(z_0) \neq 0$. It is to be emphasized that even if a function is locally univalent at *each point* of $D$, it may fail to be univalent *globally* on $D$. For example, the function $f(z) = e^z$ is locally univalent at all the points of the disc $\mathbb{D}_r := \{z : |z| < r\}$ with $r > \pi$, but fails to be globally univalent. From now on, by univalent functions, we will always mean globally univalent functions.

We will be primarily concerned with the class $\mathcal{S}$ of univalent functions on the unit disc $\mathbb{D} = \{z : |z| < 1\}$, normalized so that $f(0) = 0$ and $f'(0) = 1$. These functions are also called *schlicht*[4,5] functions. Thus each $f \in \mathcal{S}$ has a Taylor series representation of the form

$$f(z) = z + \sum_{p=2}^{\infty} b_p z^p, \qquad |z| < 1. \tag{2.1}$$

Note that a schlicht function $f(z)$ can always be obtained from an arbitrary univalent function defined on $\mathbb{D}$, $g(z)$, by an affine transformation with the definition

$$f(z) := \frac{g(z) - g(0)}{g'(0)}. \tag{2.2}$$

The usefulness of a schlicht function is that its characteristic normalization makes various numerical estimates pertaining to it simpler compared to an arbitrary univalent function.

The class $\mathcal{S}$ is preserved under a number of transformations. We will mention two of these transformations.

(i) **Conjugation:** If $f(z)$ belongs to $\mathcal{S}$, so does

$$g(z) = g(z^*)^* = z + \sum_{p=2}^{\infty} b_p^* z^p. \tag{2.3}$$

(ii) **Rotation:** The *rotation of a function $f$* is defined by

$$f_\theta(z) := e^{-i\theta} f(e^{i\theta} z), \qquad \theta \in \mathbb{R} \tag{2.4}$$

If $f \in \mathcal{S}$ then $f_\theta \in \mathcal{S}$ as well for every $\theta \in \mathbb{R}$.

**Koebe Function:** The leading example of a *schlicht* function is the *Koebe function*

$$k(z) := \frac{z}{(1-z)^2} = z + \sum_{p=2}^{\infty} p\, z^p. \tag{2.5}$$

---

[2]A domain is defined to be a simply-connected open subset of the complex plane $\mathbb{C}$.

[3]The definition of univalent function can be extended to consider meromorphic functions as well. A meromorphic univalent function on a domain $D$ can have at most one simple pole there. See [11] for a discussion. Since we are only concerned with holomorphic function in the present work, we decide to include the requirement of holomorphicity in the definition of the univalent function.

[4]In literature often, the terms univalent and schlicht are used interchangeably. Conway [12] reserves the term schlicht for describing univalent functions with the specific normalization introduced thus defining the class of schlicht functions, $\mathcal{S}$, as a subclass all the univalent functions. We follow this custom in the present work.

[5]The word "schlicht" is German and means "simple"!

Koebe function and its rotations are often solutions to various extremal problems pertaining to *schlicht* functions. Koebe function will play a central role in our subsequent analysis of scattering amplitudes.

Now that we have given a brief overview of univalent functions and the subclass of schlicht functions thereof, we will discuss a few crucial theorems and results on the schlicht functions, which will play critical roles in our analysis of the crossing-symmetric dispersive representation of scattering amplitudes.

## 2.2 Conditions for univalence of a function

In the previous section, we laid down basic notions of univalent and schlicht functions. We saw that the condition of non-vanishing first derivative of the function over a domain is not sufficient for the function to be globally univalent on the domain. However, the condition is a necessary one. In this section, we will discuss two important sufficient conditions and two necessary conditions for univalence, or equivalently schlichtness (with the specific normalization), of a function on the unit disc.

### 2.2.1 Grunsky inequalities

Grunsky inequalities [11] are necessary *and* sufficient inequalities satisfied by a function $f$ to be a schlicht on the unit disc.

Consider a holomorphic function $f : \mathbb{D} \to \mathbb{C}$ with the power series representation given by (2.1), and let

$$\ln \frac{f(t) - f(z)}{t - z} = \sum_{j,k=0}^{\infty} \omega_{j,k}\, t^j z^k, \tag{2.6}$$

with constant coefficients $\{\omega_{j,k}\}$. These are called Grunsky coefficients. It is straightforward to observe that $\omega_{j,k} = \omega_{k,j}$. These coefficients have an interesting property. Let $h : \mathbb{D} \to \mathbb{C}$ be a composition of a Mobius transformation with $f$, i.e.

$$h(z) := \frac{a f(z) + b}{c f(z) + d}, \quad ad - bc \neq 0, \tag{2.7}$$

and let $\{\widetilde{\omega}_{j,k}\}$ be corresponding Grunsky coefficients. Then,

$$\widetilde{\omega}_{j,k} = \omega_{j,k}, \quad \forall\, j, k \geq 1. \tag{2.8}$$

**Theorem 2.1.** *$f \in \mathcal{S}$ if and only if the corresponding Grunsky coefficients satisfy the inequalities*

$$\left| \sum_{j,k=1}^{N} \omega_{j,k} \lambda_j \lambda_k \right| \leq \sum_{k=1}^{N} \frac{1}{k} |\lambda_k|^2 \tag{2.9}$$

*for every positive integer $N$ and all $\lambda_k$, $k = 1, \ldots, N$.*

As an example, the Grunsky coefficients of the Koebe function $k(z)$ are given by $\omega_{j,0} = \omega_{0,j} = 2/j$, $\omega_{j,k} = -\delta_{j,k}/j$, with $\delta_{j,k}$ being the usual Kronecker delta.

**Logarithmic coefficients:** Note that the Grunsky coefficients $\{\omega_{j,0}\}$ do not enter the inequality above. One wonders whether there exists any bounding relations satisfied by these Grunsky coefficients. Indeed they satisfy very important and interesting bounds. In the literature, these Grunsky

coefficients are studied as *logarithmic coefficients* because of the simple observation

$$\ln \frac{f(z)}{z} = \sum_{n=0}^{\infty} \omega_{n,0} \, z^n, \quad f \in \mathcal{S}. \tag{2.10}$$

The *logarithmic coefficients* $\{\gamma_n\}$ are defined by

$$\gamma_n = \omega_{n,0}/2. \tag{2.11}$$

These logarithmic coefficients satisfy interesting inequalities. They satisfy the celebrated de Branges's inequalities (previously Milin conjecture ) [2] which state that for $f \in \mathcal{S}$ the corresponding logarithmic coefficients satisfy

$$\sum_{k=1}^{n} k(n-k+1) \, |\gamma_k|^2 \le \sum_{k=1}^{n} \frac{n+1-k}{k}, \qquad n = 1, 2, \ldots, \tag{2.12}$$

the equality is satisfied if and only if $f$ is a rotation of Koebe function. de Branges used this inequality in his proof of the famed Bieberbach conjecture [see section 2.4 ]. There have been attempts at obtaining sharp bounds on the individual coefficients $|\gamma_n|$. While the following sharp estimates have been obtained [2]

$$|\gamma_1| < 1, \qquad |\gamma_2| \le \frac{1}{2}\left(1 + 2e^{-2}\right), \tag{2.13}$$

the problem of finding sharp upper bounds for $|\gamma_n|$ for $n \ge 3$ in general is still an open one. However, there are some sharp estimates for modulus of logarithmic coefficients in some subclasses of $\mathcal{S}$.

### 2.2.2 Nehari conditions

In a seminal work, Nehari [13] provided a necessary and a sufficient condition for the univalence of a function on the unit disc $\mathbb{D}$. The conditions are expressed in terms of Schwarzian derivative of the function. Schwarzian derivative of a function $f(z)$ w.r.t $z$ is defined by

$$\{f(z), z\} := \left(\frac{f''(z)}{f'(z)}\right)' - \frac{1}{2}\left(\frac{f''(z)}{f'(z)}\right)^2. \tag{2.14}$$

One advantage of these conditions are that these are independent of the normalization corresponding to the schlicht functions. Thus, these conditions work with univalent functions with generic power series

$$g(z) = b_0 + b_1 z + b_2 z^2 + \ldots. \tag{2.15}$$

This is to be contrasted with the Grunsky inequalities whose precise form requires the normalization of schlicht functions.

The conditions can be stated as following theorems.

**Theorem 2.2 (Sufficient condition).** *A function $g(z)$ holomorphic on the open disc $\mathbb{D}$ will be univalent if its Schwarzian derivative satisfies the inequality*

$$|\{g(z); z\}| \le \frac{2}{(1-|z|^2)^2}. \tag{2.16}$$

**Theorem 2.3** (**Necessary condition**). *If a holomorphic function $g(z)$ is univalent in the open disc $\mathbb{D}$ then*

$$|\{g(z); z\}| \leq \frac{6}{(1-|z|^2)^2}. \tag{2.17}$$

## 2.3 Koebe growth theorem

A very important theorem that shines in our analysis of scattering amplitude is the *Koebe Growth Theorem*. This theorem, essentially, puts upper and lower bound on the magnitude of a schlicht function $f \in \mathcal{S}$.

**Theorem 2.4.** *If $f \in \mathcal{S}$ and $|z| < 1$, then*

$$\frac{|z|}{(1+|z|)^2} \leq |f(z)| \leq \frac{|z|}{(1-|z|)^2}. \tag{2.18}$$

*One of the equalities holds at some point $z \neq 0$ if and only if $f$ is a rotation of the Koebe function.*

We want to emphasize that the bounds are the consequence of $f$ being univalent. Thus, the converse of the theorem need not be true, i.e. a function defined on the unit disc $\mathbb{D}$ with the normalization same as that of a schlicht function satisfying any one of the four bounding relations above need not be univalent.

## 2.4 de Branges's theorem

One of the most important properties of univalent functions is that its Taylor coefficients are bounded. For a schlicht function $f \in \mathcal{S}$, Bieberbach proved in 1916 [1] that the second coefficient $b_2$ in the Taylor series representation (2.1) is bounded as

$$|b_2| \leq 2, \tag{2.19}$$

with equality holding *if and only if* $f$ is a rotation of the Koebe function. In the same work, Bieberbach conjectured the following bound for the general coefficient $b_n$:

$$|b_n| \leq n, \qquad \forall\, n \geq 2; \tag{2.20}$$

with the equality holding *if and only if* $f$ is a rotation of the Koebe function. This conjecture came to be known as the famed *Bieberbach Conjecture* and resisted a rigorous proof for about seven decades until Louis de Branges proved it in 1985 [2], and the result came to be known as *de Branges's Theorem*. For completeness, let us note down the full statement of de Branges's theorem.

**Theorem 2.5.** *Let $f$ be an arbitrary schlicht function, $f \in \mathcal{S}$, with the power series representation defined by (2.1). Then the Bieberbach conjecture holds true, i.e.*

$$|b_n| \leq n, \qquad \forall\, n \geq 2; \tag{2.21}$$

*with the equality holding if and only if $f$ is a rotation of the Koebe function $k(z)$ defined in (2.5), i.e. if and only if*

$$f(z) = e^{-i\theta}k(e^{i\theta}z), \qquad \forall\,\theta \in \mathbb{R}. \tag{2.22}$$

While de Branges proved the Bieberbach conjecture in its full generality only in 1985, various special cases have been proved earlier. One particular case relevant to our QFT discussion is the

Bieberbach conjecture for schlicht functions with real Taylor coefficients, $a_n \in \mathbb{R}$. This particular case was proved independently during 1931-1933 by Dieudonné [14], Rogosinski [15], and Szász [16].

## 2.5 Partial sums of univalent functions: Szegö theorem

Consider a schlicht function $f(z)$ on the unit disc with the power series representation (2.1). The $n$th partial-sum, or $n$th section, of the function $f$, denoted by $f_n$, is defined by

$$f_n(z) := z + \sum_{k=2}^{n} b_k z^k. \qquad (2.23)$$

Now, the important question is what is the domain of univalence for the partial sum $f_n$. While that is, in general, a difficult question to answer, one can still ask as to what is the largest domain over which *any section of an arbitrary $f \in \mathcal{S}$* is univalent? Szegö [17] proved the following theorem in this aspect. See [18, §8.2, pp. 241-246] for a proof.

**Theorem 2.6** (**Szegö theorem**). *Define the numbers $\{r_n \in \mathbb{R}^+\}$ such that the $m$th section of a schlicht function $f \in \mathcal{S}$, $f_m$, is univalent in the disc $\mathbb{D}_{r_n}$ for all $m \geq n$. Then,*

$$r_1 = \frac{1}{4}, \qquad (2.24)$$

*i.e., each section remains univalent in the disc $\mathbb{D}_{1/4}$, and the number $1/4$ can't be replaced by a higher one.*

The statement of the number $1/4$ not being replaceable by a higher number needs some explanation. Consider an *arbitrary $f \in \mathcal{S}$*, and let the domain over which the $n$th section $f_n$ is univalent be $\mathcal{D}_{f_n}$. Then the above theorem tells that

$$\mathcal{D}_{f_n} \supseteq \mathbb{D}_{\frac{1}{4}} \qquad \forall n \geq 2. \qquad (2.25)$$

Equivalently, this can be expressed as

$$\bigcap_{\substack{f \in \mathcal{S} \\ n \in \mathbb{Z}^+, \, n \geq 2}} \mathcal{D}_{f_n} = \mathbb{D}_{\frac{1}{4}}. \qquad (2.26)$$

The number $1/4$ is the best estimate because the domain of univalence for the second section of the Koebe function $k(z)$ is exactly equal to the disc of radius $1/4$, i.e.

$$\mathcal{D}_{k_2} = \mathbb{D}_{\frac{1}{4}}. \qquad (2.27)$$

Note that this theorem does not tell anything about the exact domains of univalence of sections of arbitrary schlicht functions. All this theorem tells us that whatever the domain of univalence of a section of a schlicht function be, it is at least large enough to contain the disc $\mathbb{D}_{\frac{1}{4}}$, or equivalently, every section of any schlicht function is univalent on $\mathbb{D}_{\frac{1}{4}}$.

The evaluation of exact domains of univalence of sections of arbitrary schlicht function is still an open problem.

# 3 Crossing symmetric dispersion relation: A brief review

We will begin our QFT discussion by reviewing key aspects of crossing symmetric dispersion relations. At the onset, we should point out that unlike the Bieberbach conjecture, where univalence played a crucial role, in QFTs, we will be able to derive certain inequalities by just using the univalence of the kernel in the dispersion relation and unitarity. This enables us to derive inequalities like $|b_n| \leq n$ as well as the two-sided bounds on $|\mathcal{M}|$ without needing univalence of the full amplitude. The question of univalence of the full scattering amplitude is a harder one to tackle in generality. We will begin a preliminary attack on this question without being able to settle the issue completely.

Scattering amplitudes of 2-2 identical scalars are functions of Mandelstam invariants $s$, $t$, $u$, they are related via $s + t + u = 4m^2 = \mu$. For our convenience, we will work with shifted variables $s_1 = s - \frac{\mu}{3}$, $s_2 = t - \frac{\mu}{3}$, $s_3 = u - \frac{\mu}{3}$. Such scattering amplitudes $\mathcal{M}(s_1, s_2)$ are fully crossing symmetric, namely $\mathcal{M}(s_1, s_2) = \mathcal{M}(s_2, s_3) = \mathcal{M}(s_3, s_1)$. Scattering amplitudes have physical branch cuts for $s_k \geq \frac{2\mu}{3}$. To write down a crossing symmetric dispersion relation the most useful trick is to parametrize the $s_1, s_2$ as [3, 4]

$$s_k = a - \frac{a\,(z - z_k)^3}{z^3 - 1}, \quad k = 1, 2, 3 \tag{3.1}$$

with $a$ being real, and $z_k$'s are cube roots of unity. The $z, a$ are crossing symmetric variables. They are related to the crossing symmetric combinations of $s_1, s_2$, namely $x = -(s_1 s_2 + s_2 s_3 + s_3 s_1) = \frac{-27a^2 z^3}{(z^3-1)^2}$, $y = -s_1 s_2 s_3 = \frac{-27a^3 z^3}{(z^3-1)^2}$, $a = y/x$.

Fully crossing symmetric amplitude can be expanded like

$$\mathcal{M}(s_1, s_2) = \sum_{p=0, q=0}^{\infty} \mathcal{W}_{p,q} x^p y^q. \tag{3.2}$$

The parametrization in (3.1), maps the physical cuts $s_k \geq \frac{2\mu}{3}$ in a unit circle in $z$-plane, see figure (1a) for $-\frac{2\mu}{9} < a < 0$. If $a < -2\mu/9$, then as [3] show, there will be branch cuts on the real $z$ axis. In the transformed variables the amplitude becomes a function of $z, a$, namely $\mathcal{M}(z, a)$. The most usefulness of (3.1) is that it enables us to write a dispersion relation which is manifestly crossing symmetric, see [3, 4].

$$\mathcal{M}(\tilde{z}, a) = \alpha_0 + \frac{1}{\pi} \int_{\frac{2\mu}{3}}^{\infty} \frac{ds_1'}{s_1'} \mathcal{A}\left(s_1'; s_2^{(+)}\left(s_1', a\right)\right) H\left(s_1', \tilde{z}\right) \tag{3.3}$$

where $\mathcal{A}(s_1; s_2)$ is the s-channel discontinuity (discontinuity of the amplitude cross $s_1 \geq \frac{2\mu}{3}$), $\alpha_0 = \mathcal{M}(z = 0, a)$ is the subtraction constant independent of $a$, and

$$\begin{aligned}
H(s_1', \tilde{z}) &= \frac{27a^2 \tilde{z}\,(2s_1' - 3a)}{27a^3 \tilde{z} - 27a^2 \tilde{z} s_1' - (1 - \tilde{z})^2 (s_1')^3}, \\
s_2^{(+)}\left(s_1', a\right) &= -\frac{s_1'}{2}\left[1 - \left(\frac{s_1' + 3a}{s_1' - a}\right)^{1/2}\right],
\end{aligned} \tag{3.4}$$

where we have introduced the new variable $\tilde{z} := z^3$. This is because all the manifestly crossing symmetric functions are functions of $\tilde{z} = z^3$.

Let us expound a bit on the analyticity structure of the amplitude on the complex $\tilde{z}-$plane. The figure (1b) below shows the image of the physical cuts in $\tilde{z} = z^3$-plane (for $-\frac{2\mu}{9} < a < 0$). Notice that in $\tilde{z} = z^3$ plane, the images of the physical cuts in all three channels are same. We will focus on the situation where $a$ is real and note that $|\tilde{z}| = 1$ if $s_1, s_2$ are real (we will set $\mu = 4m^2 = 4$ here). In the $\tilde{z}$ plane, the forward limit ($s_2 = -4/3, s_1 \geq 8/3$) corresponds to arcs that start at $\tilde{z} = -1$ and approaching $\tilde{z} = 1$ along $|\tilde{z}| = 1$. If $s_2 > -4/3$ then the full boundary of the disc is not traversed while if $s_2 \leq -4/3$ then the full boundary is traversed[6]. A further important point to keep in mind is that since real $s_1, s_2$ correspond to $|\tilde{z}| = 1$, to access the inside of the disc we need to consider complex $s_1, s_2$. Since later on, we will keep $a$ real, a complex $s_1$ will give us a complex $s_2$ since $a = s_1 s_2 (s_1 + s_2)/(s_1^2 + s_1 s_2 + s_2^2)$. Plugging back into $\tilde{z}$, we get two values, one which lies inside the disc and the other which lies outside.

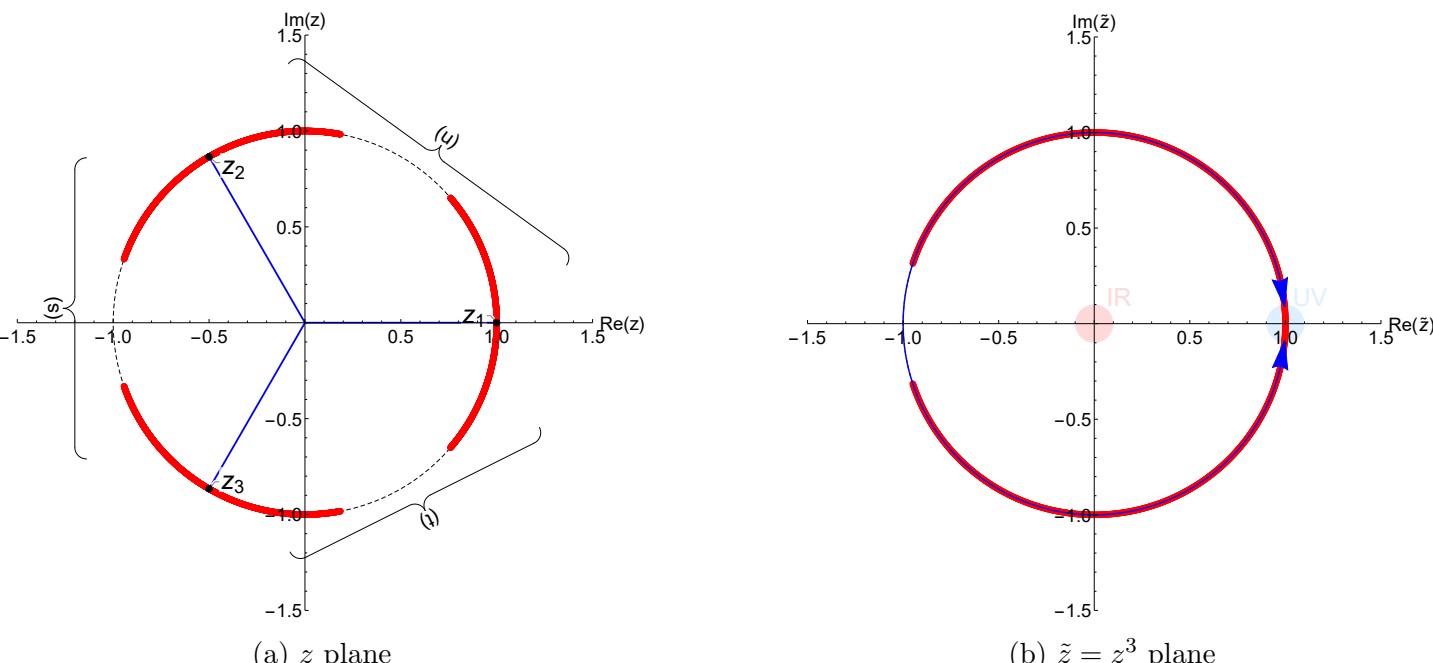

(a) $z$ plane  (b) $\tilde{z} = z^3$ plane

Figure 1: Image of the physical cuts. The blue line on $\tilde{z} = 1$ indicates the forward limit $s_2 = -4/3, s_1 \geq 8/3$. The two trajectories start from $\tilde{z} = -1$ and as $s_1$ increases they approach $\tilde{z} = 1$.

The scattering amplitude $\mathcal{M}$ admits a power series expansion about $\tilde{z} = 0$ converging in the unit disc $|\tilde{z}| < 1$,

$$\mathcal{M}(\tilde{z}, a) = \sum_{n=0}^{\infty} \alpha_n(a) a^{2n} \tilde{z}^n . \tag{3.5}$$

For a local theory[7], $\alpha_n(a) a^{2n}$ can be a polynomial in $a$ of order at most $3n$. It can be seen from the expression

$$\alpha_p(a) a^{2p} = \sum_{n=0}^{p} \sum_{m=0}^{n} \mathcal{W}_{n-m,m} a^m (-1)^{p-n} (-27)^n \ a^{2n} \binom{-2n}{p-n}. \tag{3.7}$$

---

[6]There are two trajectories corresponding to the two roots of $\tilde{z}$ which are obtained on starting with $x, y$ in terms of $\tilde{z}, a$ and solving for the latter in terms of $s_1, s_2$. If $s_2 > -4/3$ then the starting point is on the circle away from $\tilde{z} = -1$. As $s_1$ increases from $8/3$ the trajectory reaches $\tilde{z} = -1$ and then retraces along the boundary till it reaches $\tilde{z} = 1$.

[7]This follows from the expansion in (3.2), see [3, 4]

$$\mathcal{M}(z, a) = \sum_{n=0}^{\infty} \bar{w}_n(a) x^n . \tag{3.6}$$

This expression also implies that $\alpha_n(a)$ is in general a Laurent polynomial[8]. Similarly, the crossing-symmetric kernel $H(s_1', \tilde{z})$ admits a power series expansion abut $\tilde{z} = 0$:

$$H(s_1', \tilde{z}) = \sum_{n=0}^{\infty} \beta_n\left(a, s_1'\right) \tilde{z}^n, \tag{3.9}$$

with

$$\beta_n\left(a, s_1'\right) = \frac{3\sqrt{3}a2^{-n}\left(s_1'\right)^{-3n}}{\sqrt{a - s_1'}\sqrt{3a + s_1'}} \left[ \left(27a^3 - 27a^2 s_1' + 3\sqrt{3}a\sqrt{a - s_1'}\sqrt{3a + s_1'}\left(3a - 2s_1'\right) + 2\left(s_1'\right)^3\right)^n \right.$$
$$\left. - \left(27a^3 - 27a^2 s_1' + 3\sqrt{3}a\sqrt{a - s_1'}\sqrt{3a + s_1'}\left(2s_1' - 3a\right) + 2\left(s_1'\right)^3\right)^n \right]. \tag{3.10}$$

We can make two immediate and important observations from the above expression:

(I)
$$\beta_0(a, s_1) = 0 \quad \text{identically.} \tag{3.11}$$

(II)
$$\beta_1(a, s_1) = \frac{27a^2}{s_1^3}\left(3a - 2s_1\right). \tag{3.12}$$

Now, recall that, the analytic[9] domains for $a$ and $s_1$ are given by $[-2\mu/9, 2\mu/3)$ and $[2\mu/3, \infty)$. Then, one can readily infer that $\beta_1(a, s_1)$ non-vanishing for the entire physical range of $s_1$ *if and only if*[10]

$$a \in \left(-\frac{2\mu}{9}, 0\right) \cup \left(0, \frac{4\mu}{9}\right). \tag{3.13}$$

Further, in this domain of $a$, $\beta_1 < 0$ for the entire physical domain of $s_1$. This sign of $\beta_1$ will play a crucial role for various proofs in the following analysis. For the string amplitude, that we will frequently consider, $\mu = 0$. We have subtracted the massless pole, and the lower limit of the dispersion integral starts at $s_1' = 1$, which is the location of the first massive string pole. This effectively leads to the replacement $\mu \to 3/2$ in the above discussion given $a \in \left[-\frac{1}{3}, 0\right) \cup \left(0, \frac{2}{3}\right)$.

The coefficients $\{\beta_n(a, s_1)\}$ are of extreme importance because using the crossing symmetric dispersion relation (3.3) along with (3.9), we can write an inversion formula

$$a^{2n}\alpha_n(a) = \frac{1}{\pi}\int_{\frac{2\mu}{3}}^{\infty}\frac{ds_1'}{s_1'}\,\mathcal{A}\left(s_1'; s_2^{(+)}(s_1', a)\right)\beta_n\left(a, s_1'\right), \quad n > 0. \tag{3.14}$$

Thus, we see that the $\alpha_n$s are essentially integral transforms of $\beta_n$s convoluted with the $s-$channel absorptive part $\mathcal{A}\left(s_1; s_2^{(+)}(s_1, a)\right)$.

Let us conclude this section with a significant result on the absorptive part, which will be crucial

---

[8]A Laurent polynomial $\ell(x)$ over a field $\mathbb{F}$ is an expression of the form

$$\ell(x) = \sum_{k \in \mathbb{Z}} \delta_k\, x^k, \qquad \delta_k \in \mathbb{F} \tag{3.8}$$

where now $k$ need not be necessarily positive and only finitely many coefficients $\delta_k$ are non-zero.

[9]We are calling the above domain of $a$ to be "analytic" since for this domain of $a$, the branch cuts in the complex $\tilde{z}$ plane do not lie along the real line.

[10]The $a = 0$ point is trivial since both $x, y = 0$ and the amplitude is a constant. In what follows, if on occasion we imply $a \neq 0$, this is to be kept in mind.

for our subsequent analysis in light of the inversion formula above.

**Lemma 3.1** (**Positivity lemma**). For a *unitary* theory, if $a \in \left(-\frac{2\mu}{9}, \frac{2\mu}{3}\right)$ then the absorptive part of the amplitude, $\mathcal{A}\left(s_1; s_2^{(+)}(s_1, a)\right)$, is *non-negative* for $s_1 \in \left[\frac{2\mu}{3}, \infty\right)$.

**Proof.** The $s$-channel discontinuity has a partial wave expansion

$$\mathcal{A}\left(s_1; s_2^{(+)}(s_1, a)\right) = \Phi(s_1; \alpha) \sum_{\ell=0}^{\infty} (2\ell + 2\alpha) a_\ell(s_1) C_\ell^{(\alpha)}\left(\sqrt{\xi(s_1, a)}\right)$$

$$\xi(s_1, a) = \cos^2 \theta_s = \left(1 + \frac{2s_2^+(s_1, a) + \frac{2\mu}{3}}{s_1 - \frac{2\mu}{3}}\right)^2 = \xi_0 + 4\xi_0\left(\frac{a}{s_1 - a}\right) \tag{3.15}$$

with $\xi_0 = \frac{s_1^2}{(s_1 - 2\mu/3)^2}$ and $\alpha = \frac{d-3}{2}$. Over the domain of $s_1 \in \left[\frac{2\mu}{3}, \infty\right)$, we find that, $\sqrt{\xi(s_1, a)} \geq 1$, which implies $C_\ell^{(\alpha)}\left(\sqrt{\xi(s_1, a)}\right) > 0$ if $a \in \left(-\frac{2\mu}{9}, \frac{2\mu}{3}\right)$. Next, for the given domains of $s_1$ and $a$ one has $\Re\left[s_2^+(s_1, a)\right] \in \left[-\frac{\mu}{3}, \frac{2\mu}{3}\right]$. Now the analyticity domain $E(s_1)$ of $\mathcal{A}(s_1, s_2^+)$ in $t$ has been determined [19] to be

$$E(s_1) = \begin{cases} E\left(0, \frac{2\mu}{3} - s_1 \,\middle|\, 4\mu + \frac{48\mu}{3s_1 - 2\mu}\right), & \frac{2\mu}{3} < s_1 < \frac{11\mu}{3} \\[3ex] E\left(0, \frac{2\mu}{3} - s_1 \,\middle|\, \frac{192\mu}{3s_1 + \mu}\right), & \frac{11\mu}{3} < s_1 < \frac{23\mu}{3} \\[3ex] E\left(0, \frac{2\mu}{3} - s_1 \,\middle|\, \mu + \frac{48\mu}{3s_1 - 11\mu}\right), & s_1 > \frac{23\mu}{3} \end{cases} \tag{3.16}$$

where $E(f_1, f_2 | d)$ stands for an ellipse with foci at $s_2^+ = f_1$, $s_2^+ = f_2$ and right extremity at $s_2^+ = d$. It is straightforward to see that our $s_2^+$ values always lie in the interior of $E(s_1)$, i.e. the partial wave expansion for $\mathcal{A}\left(s_1; s_2^{(+)}(s_1, a)\right)$ above converges for the given domains of $a$ and $s_1$. Next, $0 \leq a_\ell(s_1) \leq 1$ on $s_1 \in \left[\frac{2\mu}{3}, \infty\right)$ as a consequence of unitarity. Therefore, if $a \in \left(-\frac{2\mu}{9}, \frac{2\mu}{3}\right)$, $\mathcal{A}\left(s_1; s_2^{(+)}(s_1, a)\right)$, is *non-negative* for $s_1 \in \left[\frac{2\mu}{3}, \infty\right)$. If $\mu = 0$, where $\xi = 1 + 4\frac{a}{a - s_1}$, for $\xi > 1$, $a > 0$ must hold[11]. $\qquad\square$

# 4 Bounds on $\{\alpha_n(a)\}$

We can bound the Taylor coefficients $\{\alpha_n(a)\}$ appearing in the power-series representation of the scattering amplitude $\mathcal{M}$, (3.5). Towards that end, let us first prove a lemma that will be used repeatedly in our analysis that follows.

**Lemma 4.1.** Consider the kernel $H(\tilde{z}; s_1, a)$ of the dispersion relation given by (3.4),

$$H(\tilde{z}; s_1, a) = \frac{27a^2\tilde{z}(2s_1 - 3a)}{27a^3\tilde{z} - 27a^2\tilde{z}s_1 - (\tilde{z} - 1)^2 (s_1)^3}. \tag{4.1}$$

Define the function

$$F(\tilde{z}; s_1, a) := \frac{H(\tilde{z}; s_1, a)}{\beta_1(a, s_1)}. \tag{4.2}$$

---

[11] For the string case, however, we will find that for $a < 0$ the bounds we will consider will still hold. We do not have a general explanation for this apart from observing that $\alpha_1 < 0$ for certain $-1/3 < a < 2/3$, which is the range of $a$ we will be interested in.

For $a \in \left(-\frac{2\mu}{9}, 0\right) \cup \left(0, \frac{4\mu}{9}\right)$ and $s_1 \in \left[\frac{2\mu}{3}, \infty\right)$, $F(\tilde{z}; s_1, a)$ is a schlicht function, or equivalently, $H(\tilde{z}; s_1, a)$ is a *univalent function on the unit disc* $|\tilde{z}| < 1$.

**Proof .** For $a \in \left(-\frac{2\mu}{9}, 0\right) \cup \left(0, \frac{4\mu}{9}\right)$ and $s_1 \in \left[\frac{2\mu}{3}, \infty\right)$, we have already proved that $\beta_1(a, s_1) \neq 0$. Thus, the function $F$ is well-defined in these domains of $a$ and $s_1$. Further, since $\beta_0 = 0$ identically, $F(\tilde{z})$ admits a power series expansion about $\tilde{z} = 0$:

$$F(\tilde{z}; s_1, a) = \tilde{z} + \sum_{n=2}^{\infty} \frac{\beta_n(a, s_1)}{\beta_1(a, s_1)} \tilde{z}^n. \tag{4.3}$$

First note that

$$F(\tilde{z}; s_1, a) = \frac{\tilde{z}}{1 + \gamma \tilde{z} + \tilde{z}^2}$$

with $\gamma = 27 \left(\frac{a}{s_1}\right)^2 \left(1 - \frac{a}{s_1}\right) - 2$. To avoid a singularity inside the unit disc we need

$$|\gamma| < 2,$$

which translates to $a \in \left(-\frac{2\mu}{9}, 0\right) \cup \left(0, \frac{4\mu}{9}\right)$ for real $a$, which is the same condition mentioned above[12].

Next, observe that we can write

$$F(\tilde{z}; s_1, a) = k(\tilde{z}) \left[1 - \frac{27a^2(a - s_1)}{s_1^3} k(\tilde{z})\right]^{-1}, \tag{4.4}$$

where $k(\tilde{z})$ is the Koebe function defined in (2.5). It is straightforward to see that $F$ can be considered as a composition of a Moebius transformation with the Koebe function. Then, by (2.8), $F$ has the Grunsky coefficients

$$\omega_{p,q} = -\frac{\delta_{p,q}}{p}, \quad p, q \geq 1, \tag{4.5}$$

and these satisfy the Grunsky inequalities of (2.9) for all $N \geq 1$. Since Grunsky inequalities are necessary and sufficient for an analytic function inside the unit disc to be univalent, this completes the proof. $\square$

Observe that, $F(\tilde{z}; s_1, a)$ and $H(\tilde{z}; s_1, a)$ are related by an affine transformation. Thus, the schlichtness of $F$ implies that $H$ *is an univalent function on the unit disc* $|\tilde{z}| < 1$ for the same domains of $a$ and $s_1$.

**Corollary 4.1.1.** For $a \in \left(-\frac{2\mu}{9}, 0\right) \cup \left(0, \frac{4\mu}{9}\right)$ and $s_1 \in \left[\frac{2\mu}{3}, \infty\right)$, the Taylor coefficients $\{\beta_n(a, s_1)\}$ in the power-series expansion of $H$ are bounded by

$$\left|\frac{\beta_n(a, s_1)}{\beta_1(a, s_1)}\right| \leq n, \quad n \geq 2 \tag{4.6}$$

*Proof.* Since $F(\tilde{z}; s_1, a)$ is a Schllicht function on the unit disc $|\tilde{z}| < 1$ in the given domains of $a$ and $s_1$, we can apply *de Branges's theorem* to the same to obtain the bound. $\square$

Let us note down another corollary of the lemma 4.1 for future reference.

**Corollary 4.1.2.** For $a \in \left(-\frac{2\mu}{9}, 0\right) \cup \left(0, \frac{4\mu}{9}\right)$ and $s_1 \in \left[\frac{2\mu}{3}, \infty\right)$, the Taylor coefficients $\{\beta_n(a, s_1)\}$

---

[12]Of course we could consider complex $a$ as well at this stage. However, since we want to make use of the positivity of the absorptive part of the amplitude later on, we will restrict our attention to real $a$.

in the power-series expansion of $H$ are bounded by

$$\frac{|\tilde{z}|}{(1+|\tilde{z}|)^2} \leq |F(\tilde{z}; s_1, a)| \leq \frac{|\tilde{z}|}{(1-|\tilde{z}|)^2}, \quad |\tilde{z}| < 1. \tag{4.7}$$

*Proof.* Applying Koebe growth theorem 2.4 to $F(\tilde{z}; s_1, a)$, one obtains the bounds. □

Now that we have collected the necessary results, let us now turn to prove the following theorem.

**Theorem 4.2.** *For non-zero $\mathcal{M}(\tilde{z}, a)$ and $a \in \left(-\frac{2\mu}{9}, 0\right) \cup \left(0, \frac{4\mu}{9}\right)$, with $\mu > 0$,*

$$\left|\frac{\alpha_n(a)a^{2n}}{\alpha_1(a)a^2}\right| \leq n, \quad \forall n \geq 2. \tag{4.8}$$

**Proof .** First, we make sure that the ratio is well-defined in $a \in \left(-\frac{2\mu}{9}, 0\right) \cup \left(0, \frac{4\mu}{9}\right)$. In particular, we need to make sure that $\alpha_1(a) \neq 0$ for $a \in \left(-\frac{2\mu}{9}, 0\right) \cup \left(0, \frac{4\mu}{9}\right)$ since $a^2$ is never zero due to $\mu \neq 0$. To do so, let us start with the inversion formula, (3.14), to write

$$-\alpha_1(a)a^2 = \frac{1}{\pi} \int_{\frac{2\mu}{3}}^{\infty} \frac{ds_1'}{s_1'} \mathcal{A}\left(s_1'; s_2^{(+)}(s_1', a)\right) \left[-\beta_1\left(a, s_1'\right)\right]. \tag{4.9}$$

From the *positivity lemma* 3.1, we have that $\mathcal{A}\left(s_1'; s_2^{(+)}(s_1', a)\right) \geq 0$ for $a \in \left(-\frac{2\mu}{9}, 0\right) \cup \left(0, \frac{4\mu}{9}\right)$ and $s_1' \in \left[\frac{2\mu}{3}, \infty\right)$. Further, we have already seen that $\beta_1\left(a, s_1'\right) < 0$ for the same domains of $a$ and $s_1'$. Thus, the integrand is *non-negative*. Further, since $\mathcal{M}(\tilde{z}, a)$ is non-zero, $\mathcal{A}\left(s_1'; s_2^{(+)}(s_1', a)\right)$ does not vanish identically over $s_1' \in \left[\frac{2\mu}{3}, \infty\right)$. Hence, the integral will be *positive* implying

$$\alpha_1(a) < 0, \quad a \in \left(-\frac{2\mu}{9}, \frac{4\mu}{9}\right). \tag{4.10}$$

Therefore, the ratio under consideration,

$$a^{2n-2} \frac{\alpha_n(a)}{\alpha_1(a)} \tag{4.11}$$

is well-defined for $a \in \left(-\frac{2\mu}{9}, 0\right) \cup \left(0, \frac{4\mu}{9}\right)$.

Now, let us use the inversion formula (3.14) once again, and taking the absolute values on both sides of the formula one obtains

$$
\begin{aligned}
\left|\alpha_n(a)a^{2n}\right| &= \frac{1}{\pi} \left|\int_{\frac{2\mu}{3}}^{\infty} \frac{ds_1'}{s_1'} \mathcal{A}\left(s_1'; s_2^{(+)}(s_1', a)\right) \beta_n\left(a, s_1'\right)\right|, \\
&\leq \frac{1}{\pi} \int_{\frac{2\mu}{3}}^{\infty} \frac{ds_1'}{s_1'} \left|\mathcal{A}\left(s_1'; s_2^{(+)}(s_1', a)\right) \beta_n\left(a, s_1'\right)\right| \quad \text{[Applying triangle inequality]}, \\
&= \frac{1}{\pi} \int_{\frac{2\mu}{3}}^{\infty} \frac{ds_1'}{s_1'} \mathcal{A}\left(s_1'; s_2^{(+)}(s_1', a)\right) \left|\beta_n\left(a, s_1'\right)\right| \quad \left[\mathcal{A}\left(s_1'; s_2^{(+)}(s_1', a)\right) \geq 0 \text{ by lemma 3.1}\right],
\end{aligned}
$$

$$\leq \frac{1}{\pi} \int_{\frac{2\mu}{3}}^{\infty} \frac{ds_1'}{s_1'} \mathcal{A}\left(s_1'; s_2^{(+)}(s_1', a)\right) n \left|\beta_1\left(a, s_1'\right)\right| \quad \text{[Applying corollary 4.1.1]},$$

$$= \frac{n}{\pi} \int_{\frac{2\mu}{3}}^{\infty} \frac{ds_1'}{s_1'} \mathcal{A}\left(s_1'; s_2^{(+)}(s_1', a)\right) \left[-\beta_1\left(a, s_1'\right)\right] \quad \left[\beta_1 < 0 \iff |\beta_1| = -\beta_1\right],$$

$$= n\left(-\alpha_1(a)a^2\right) = n\left|\alpha_1(a)a^2\right| \quad \left[\alpha_1(a) < 0 \text{ from } (4.10)\right]. \tag{4.12}$$

$\therefore$ For $a \in \left(-\frac{2\mu}{9}, 0\right) \cup \left(0, \frac{4\mu}{9}\right)$,

$$\left|\frac{\alpha_n(a)a^{2n}}{\alpha_1(a)a^2}\right| \leq n, \qquad \forall\, n \geq 2. \tag{4.13}$$

$\square$

For illustration, we show $\left|\frac{\alpha_n(a)a^{2n}}{\alpha_1(a)a^2}\right|$ as a function of $a$ for 1-loop $\phi^4$-amplitude and tree level type II string amplitude in figure (2). The presented proof assumes the range $a \in \left(-\frac{2\mu}{9}, 0\right) \cup \left(0, \frac{4\mu}{9}\right)$, even though the plots suggest that bound is still valid till $a \in \left(-\frac{2\mu}{9}, 0\right) \cup \left(0, \frac{2\mu}{3}\right)$, for some cases. For the string amplitude $a \in \left(-\frac{s_1^{(0)}}{3}, 0\right) \cup \left(0, \frac{2s_1^{(0)}}{3}\right)$, where $s_1^{(0)}$ is the starting point of the physical cuts in s-channel. We use the fact that first massive state is at $s_1^{(0)} = 1$.

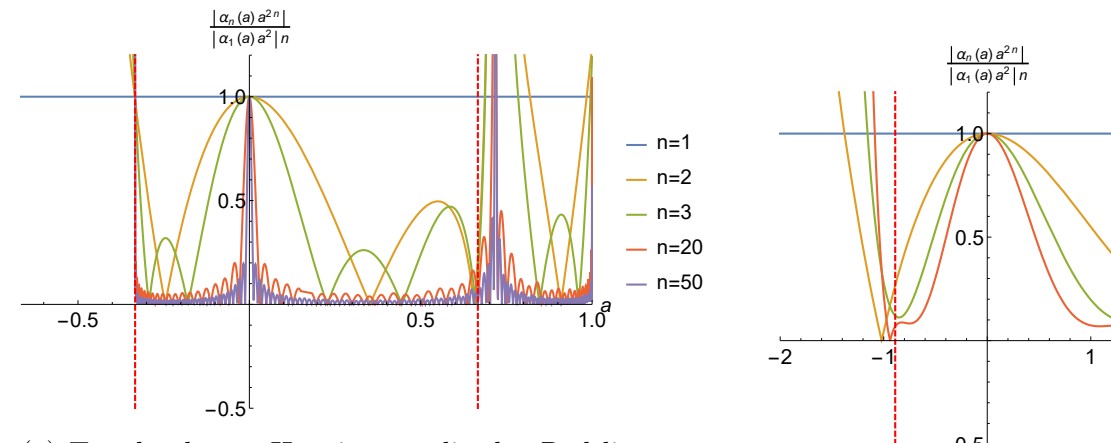

(a) Tree level type II string amplitude. Red lines are $a = -\frac{1}{3}, \frac{2}{3}$

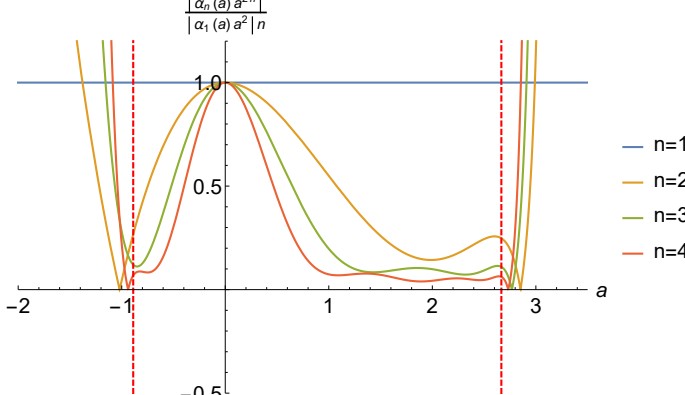

(b) 1-loop $\phi^4$-amplitude. Red lines are $a = -\frac{8}{9}, \frac{8}{3}$

Figure 2: Bounds on $\left|\frac{\alpha_n(a)a^{2n}}{\alpha_1(a)a^2}\right|$ as a function of $a$

## 5 Stronger bounds on the Wilson coefficients $\mathcal{W}_{p,q}$

In order to derive bounds on $\mathcal{W}_{p,q}$, we first recall the formula of (3.7). We have already proved in equation (4.10) that $\alpha_1 < 0$ for $a \in \left(-\frac{2\mu}{9}, 0\right) \cup \left(0, \frac{4\mu}{9}\right)$. Now, $\alpha_1 = -\mathcal{W}_{1,0}\left(a\frac{\mathcal{W}_{0,1}}{\mathcal{W}_{1,0}} + 1\right)$ which follows from (3.7). Further, $\mathcal{W}_{1,0}$ is strictly positive, which was shown in [4, 9, 23]. This immediately implies that $\left(a\frac{\mathcal{W}_{0,1}}{\mathcal{W}_{10}} + 1\right) > 0$ for the given range of $a$ quoted above. From here, the bound in (5.1) follows.

$$-\frac{9}{4\mu} < \frac{\mathcal{W}_{0,1}}{\mathcal{W}_{1,0}} < \frac{9}{2\mu}. \tag{5.1}$$

It is to be emphasized that if (5.1) is not satisfied, i.e., if $\mathcal{W}_{0,1}/\mathcal{W}_{1,0}$ is outside the range given by (5.1), then dialing $a$ within the range $-\frac{2\mu}{9} < a < \frac{4\mu}{9}$, $a \neq 0$, one can make the factor $\left(a\frac{\mathcal{W}_{0,1}}{\mathcal{W}_{1,0}} + 1\right)$

*change sign* contradicting $\alpha_1(a) < 0$ in the said range of $a$.[13] Let us present this derivation a little differently as well. Using theorem 4.2 for $n = 2$ and (3.7), we find

$$-2 \le 2 - \frac{27a^2 \left(a \left(a\mathcal{W}_{0,2} + \mathcal{W}_{1,1}\right) + \mathcal{W}_{2,0}\right)}{a\mathcal{W}_{0,1} + \mathcal{W}_{1,0}} \le 2 \,. \tag{5.2}$$

Now if the denominator $a\mathcal{W}_{0,1} + \mathcal{W}_{1,0}$ vanishes, then unless at the same point the numerator vanishes, we will contradict the inequality. Suppose for $a = a_0$, the denominator vanishes. Then we must have $a_0(a_0\mathcal{W}_{0,2} + \mathcal{W}_{1,1}) + \mathcal{W}_{2,0} = 0$ giving a relation between three apparently independent Wilson coefficients. This appears unnatural to us and if we were to avoid this possibility we would again get eq.(5.1). Using eq.(5.2), it is also possible to deduce bounds [27] on the individual $\mathcal{W}_{0,2}/\mathcal{W}_{1,0}, \mathcal{W}_{1,1}/\mathcal{W}_{1,0}, \mathcal{W}_{2,0}/\mathcal{W}_{1,0}$ and these appear comparable to [10].

The condition $-\frac{9}{4\mu} < \frac{\mathcal{W}_{0,1}}{\mathcal{W}_{1,0}}$ was derived in [4, eq (5)], while $\frac{\mathcal{W}_{0,1}}{\mathcal{W}_{1,0}} < \frac{9}{2\mu}$ is a new finding. For illustration purpose, we show in figure (3) that pion S-matrices satisfy them in a very non-trivial way. We will comment more on the behaviour exhibited in figure (3) below.

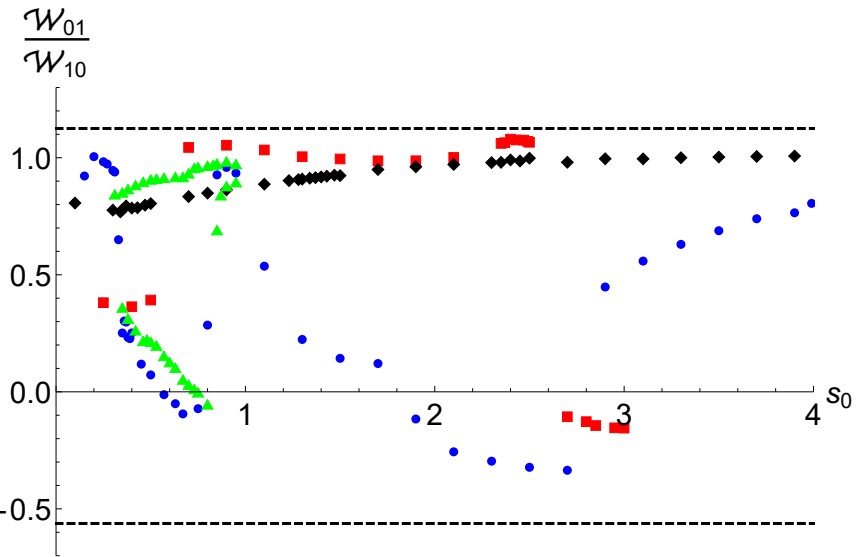

Figure 3: Ratio of $\frac{\mathcal{W}_{0,1}}{\mathcal{W}_{1,0}}$ obtained from the S-matrix bootstrap. The horizontal axis is the Adler zero $s_0$. The green points are for the pion lake [20]. The blue and red points are for the upper and lower river boundaries [21, 22] while the black points are for the line of minimum averaged total cross section S-matrices [22].

For the string example, we can ask the following question: Given $\mathcal{W}_{0,1}, \mathcal{W}_{1,0}, \mathcal{W}_{1,1}, \mathcal{W}_{2,0}$, in other words the Wilson coefficients till the eight-derivative order term $x^2$, how constraining is (5.2)? The situation is shown in fig(4a). Quite remarkably, the range of $\mathcal{W}_{02}$ is very limited; the inequality is very constraining indeed!

We can further investigate the situation for $n = 3$. We get

$$-3 \le 3 + \frac{27a^2 \left(a \left(-4a\mathcal{W}_{0,2} + 27a \left(a \left(a \left(a\mathcal{W}_{0,3} + \mathcal{W}_{1,2}\right) + \mathcal{W}_{2,1}\right) + \mathcal{W}_{3,0}\right) - 4\mathcal{W}_{1,1}\right) - 4\mathcal{W}_{2,0}\right)}{a\mathcal{W}_{0,1} + \mathcal{W}_{1,0}} \le 3 \tag{5.3}$$

We give in the appendix a demonstration of how to constrain $\mathcal{W}_{0,3}$ using this inequality.

---

[13]Note that if we were to find stronger bounds on the ratio, we would need a *wider* allowed range for $a$ and vice versa, contrary to the naive expectation.

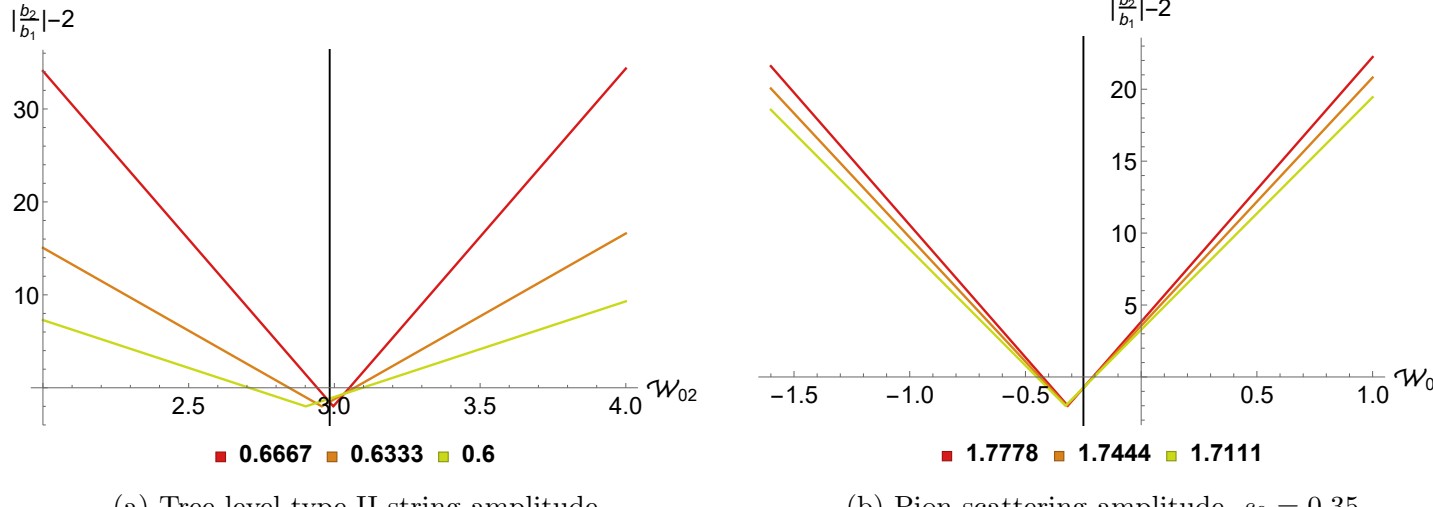

(a) Tree level type II string amplitude

(b) Pion scattering amplitude, $s_0 = 0.35$

Figure 4: Constraints on Wilson coefficients using (5.2). Given $\mathcal{W}_{0,1}, \mathcal{W}_{1,0}, \mathcal{W}_{1,1}, \mathcal{W}_{2,0}$ figure shows that bound on the $\mathcal{W}_{0,2}$. Since $\left|\frac{b_2}{b_1}\right| - 2$ should be less than zero, $\mathcal{W}_{0,2}$ must lie inside the triangle. Black line is the exact answer. Different values of $a$ are indicated with different colours.

## 5.1    Bounds in case of EFTs:

In an EFT, usually, the Lagrangian is known up to some energy scale. From that information, one can calculate the amplitude up to that scale. In such cases, we can subtract off the known part of the amplitude. These steps result in a shift in the lower limit of the dispersion integral (3.3) by the scale $\delta_0$, namely $\mu \to \mu + 3\delta_0/2$ (see [4]). Therefore, making this replacement in (5.1), we have

$$-\frac{9}{4\mu + 6\delta_0} < \frac{\mathcal{W}_{0,1}}{\mathcal{W}_{1,0}} < \frac{9}{2\mu + 3\delta_0} \,. \tag{5.4}$$

Here $\mathcal{W}$ are the Wilson coefficients of the amplitude with subtractions. Now notice that if we consider $\delta_0 \gg \mu$ then we have

$$-\frac{3}{2\delta_0} < \frac{\mathcal{W}_{0,1}}{\mathcal{W}_{1,0}} < \frac{3}{\delta_0} \,. \tag{5.5}$$

Let us compare this to [10]. Converting their results to our conventions, we find that the lower bound above is identical to their findings–this is corroborated by the results of [9] as well as what arises from crossing symmetric dispersion relations [4]. The other side of the bound is more interesting. The strongest result in $d = 4$ in [10] places the upper bound at $\approx 5.35/\delta_0$ in our conventions. Their approach also makes the bound spacetime dimension dependent. Now remarkably, the bound we quote above and the $d \gg 1$ limit of [10] are identical! In EFT approaches, one takes the so-called null constraints or locality constraints and expands in the limit $\delta_0 \gg \mu$. It is possible that a more exact approach building on [10] will lead to a stronger bound as in (7.10).

Let us now comment on the behaviour in the figure (3). First, notice that all S-matrices appear to respect the upper bound we have found above; for the S-matrix bootstrap results, we set $\delta_0 = 0$. For comparison, note that for 1-loop $\phi^4$, and 2-loop chiral perturbation theory, we have

$$\left(\frac{\mathcal{W}_{0,1}}{\mathcal{W}_{1,0}}\right)_{\phi^4} \approx -0.315 \,, \quad \left(\frac{\mathcal{W}_{0,1}}{\mathcal{W}_{1,0}}\right)_{\chi - PT} \approx -0.135 \,, \tag{5.6}$$

both in units where $m = 1$. These numbers would be closer to the lower black dashed line in the figure (3), which is the bound that is common in all approaches so far. The upper black dashed line is

what we find in the current paper. For future work, it will be interesting to search for an interpolating bound as in (7.9) which enables us to interpolate between $\delta_0 = 0$ and $\delta_0 \gg \mu$.

# 6 Two-sided bounds on the amplitudes

The crossing symmetric dispersion relation can enable us to derive $2-$sided bounds on the scattering amplitude $\mathcal{M}$. In this section, we will derive such bounds. The result that we will first prove is the following:

**Theorem 6.1.** *Let $\mathcal{M}(\tilde{z}, a)$ be a unitary and crossing-symmetric scattering amplitude admitting the dispersive representation* (3.3) *and admits the power series expansion* (3.5) *about $\tilde{z} = 0$ which converges in the open disc $|\tilde{z}| < 1$. Define the function*

$$f(\tilde{z}, a) := \frac{\mathcal{M}(\tilde{z}, a) - \alpha_0}{\alpha_1(a) a^2}, \qquad \alpha_0 = \mathcal{M}(\tilde{z} = 0, a). \tag{6.1}$$

*Then for $a \in \left( -\frac{2\mu}{9}, 0 \right) \cup \left( 0, \frac{4\mu}{9} \right)$,*

*(1)*

$$|f(\tilde{z}, a)| \leq \frac{|\tilde{z}|}{(1 - |\tilde{z}|)^2}, \quad |\tilde{z}| < 1 \tag{6.2}$$

*(2)*

$$|f(\tilde{z}, a)| \geq \frac{|\tilde{z}|}{(1 + |\tilde{z}|)^2}, \quad \tilde{z} \in \mathbb{R} \wedge |\tilde{z}| < 1. \tag{6.3}$$

**Proof.** Let us first prove the upper bound. Starting with the dispersion relation (3.3), we can obtains

$$
\begin{aligned}
|\mathcal{M}(\tilde{z}, a) - \alpha_0| &\leq \int_{\frac{2\mu}{3}}^{\infty} \frac{ds'_1}{s'_1} \left| \mathcal{A}\left( s'_1; s_2^{(+)}(s'_1, a) \right) \right| \left| H(s'_1, \tilde{z}) \right| \quad \text{[Applying triangle inequality]}, \\
&= \frac{1}{\pi} \int_{\frac{2\mu}{3}}^{\infty} \frac{ds'_1}{s'_1} \mathcal{A}\left( s'_1; s_2^{(+)}(s'_1, a) \right) |\beta_1(a, s'_1)| \left| F(\tilde{z}; s'_1, a) \right| \quad \text{[Using lemma 3.1 and (4.2)]}, \\
&\leq \frac{1}{\pi} \int_{\frac{2\mu}{3}}^{\infty} \frac{ds'_1}{s'_1} \mathcal{A}\left( s'_1; s_2^{(+)}(s'_1, a) \right) |\beta_1(a, s'_1)| \frac{|\tilde{z}|}{(1 - |\tilde{z}|)^2} \quad \text{[Using corollary 4.7]}, \\
&= \frac{|\tilde{z}|}{(1 - |\tilde{z}|)^2} \frac{1}{\pi} \int_{\frac{2\mu}{3}}^{\infty} \frac{ds'_1}{s'_1} \mathcal{A}\left( s'_1; s_2^{(+)}(s'_1, a) \right) \left[ -\beta_1(a, s'_1) \right] \quad [\beta_1 < 0 \iff |\beta_1| = -\beta_1], \\
&= \frac{|\tilde{z}|}{(1 - |\tilde{z}|)^2} \left[ -\alpha_1(a) a^2 \right] = \frac{|\tilde{z}|}{(1 - |\tilde{z}|)^2} \left| \alpha_1(a) a^2 \right| \quad [\alpha_1(a) < 0 \text{ from (4.10)}]. \tag{6.4}
\end{aligned}
$$

Thus, we have finally,

$$\left| \frac{\mathcal{M}(\tilde{z}, a) - \alpha_0}{\alpha_1(a) a^2} \right| \leq \frac{|\tilde{z}|}{(1 - |\tilde{z}|)^2}, \quad |\tilde{z}| < 1; \tag{6.5}$$

which proves part (1) of the theorem.

Next, let us prove the lower bound. First, we write

$$F(\tilde{z}; s_1, a) \equiv \frac{H(\tilde{z}; s_1, a)}{\beta_1(a, s_1)} = \frac{\tilde{z}}{(1 + \tilde{z})^2} \times \rho(\tilde{z}; s_1, a), \tag{6.6}$$

where

$$\rho(\tilde{z}; s_1, a) := \left( \frac{1 + \tilde{z}}{1 - \tilde{z}} \right)^2 \times \left[ 1 - \frac{27 a^2 (a - s_1)}{s_1^3} k(\tilde{z}) \right]^{-1}, \tag{6.7}$$

$k(\tilde{z})$ being the Koebe function of (2.5). Now it turns out that $\rho > 1$ for $\tilde{z} \in \mathbb{R}^+$ and $\rho < -1$ for

$\tilde{z} \in \mathbb{R}^-$. This immediately implies

$$|\rho| > 1, \tag{6.8}$$

$$\tilde{z}\rho = |\tilde{z}||\rho|. \tag{6.9}$$

Next, we use the dispersion relation and (4.2) to get

$$
\begin{aligned}
|\mathcal{M}(\tilde{z}, a) - \alpha_0| &= \frac{1}{\pi} \left| \int_{\frac{2\mu}{3}}^{\infty} \frac{ds_1'}{s_1'} \mathcal{A}\left(s_1'; s_2^{(+)}(s_1', a)\right) \beta_1(a, s_1') F(\tilde{z}; s_1, a) \right|, \\
&= \frac{1}{\pi} \left| \int_{\frac{2\mu}{3}}^{\infty} \frac{ds_1'}{s_1'} \mathcal{A}\left(s_1'; s_2^{(+)}(s_1', a)\right) \beta_1(a, s_1') \frac{\tilde{z}}{(1+\tilde{z})^2} \times \rho(\tilde{z}; s_1, a) \right| \quad \text{[Using (6.6)]}, \\
&= \frac{1}{\pi} \left| \int_{\frac{2\mu}{3}}^{\infty} \frac{ds_1'}{s_1'} \mathcal{A}\left(s_1'; s_2^{(+)}(s_1', a)\right) \beta_1(a, s_1') \frac{|\tilde{z}|}{(1+\tilde{z})^2} \times |\rho(\tilde{z}; s_1, a)| \right| \quad \text{[Using (6.9)]}, \\
&\geq \frac{1}{\pi} \frac{|\tilde{z}|}{(1+|\tilde{z}|)^2} \left| \int_{\frac{2\mu}{3}}^{\infty} \frac{ds_1'}{s_1'} \mathcal{A}\left(s_1'; s_2^{(+)}(s_1', a)\right) \beta_1(a, s_1') \right| \quad \text{[Using triangle inequality and (6.8)]}, \\
&= \frac{1}{\pi} \frac{|\tilde{z}|}{(1+|\tilde{z}|)^2} \left| \alpha_1(a)a^2 \right|. \tag{6.10}
\end{aligned}
$$

Therefore, we have finally,

$$\left| \frac{\mathcal{M}(\tilde{z}, a) - \alpha_0}{\alpha_1(a)a^2} \right| \geq \frac{|\tilde{z}|}{(1+|\tilde{z}|)^2}, \quad \tilde{z} \in \mathbb{R} \wedge |\tilde{z}| < 1; \tag{6.11}$$

proving the second part and, hence, the theorem. $\qquad \square$

We emphasize that our derivation for upper bound considers $\tilde{z}$ as complex numbers, while we present the derivation for lower bound considering only $\tilde{z}$ real numbers. We worked with a range of $-\frac{2\mu}{9} < a < \frac{4\mu}{9}$. Nevertheless, both of the bounds are valid for complex $\tilde{z}$ as far as we have observed. These bounds are satisfied by the $1-$loop $\phi^4$ amplitude and close string amplitude even for complex $\tilde{z}$, demonstrated in figure (5), (6).

Even though, we have presented our proof for the range $a \in \left(-\frac{2\mu}{9}, 0\right) \cup \left(0, \frac{4\mu}{9}\right)$, for certain cases, we observed that bound is still valid till $a \in \left(-\frac{2\mu}{9}, 0\right) \cup \left(0, \frac{2\mu}{3}\right)$. For massless amplitude its $a \in \left(-\frac{s_1^{(0)}}{3}, 0\right) \cup \left(0, \frac{2s_1^{(0)}}{3}\right)$, where $s_1^{(0)}$ is the starting point of the physical cuts in s-channel. Let us now rewrite the two sided bounds of the familiar Mandelstam variables. Now first note that, for real $\tilde{z} > 0$, in terms of $x, a$ variables, the bounds become

$$\frac{a^2|\alpha_1(a)||x|}{4|x| + 27a^2} \leq |\mathcal{M}(z, a) - \alpha_0| \leq \frac{|\alpha_1(a)||x|}{27}. \tag{6.12}$$

Here since $x = -27a^2\tilde{z}/(\tilde{z} - 1)^2$, $x$ is real and negative. Now let us examine these bounds in the Regge limit where $|s_1| \to \infty$ with $s_2$ fixed. From (3.1), it is clear that in this case in the $z$-variable, $z \to e^{2\pi i/3}$ which also takes $s_2 \to a$. Now writing $s_1 = |s_1|e^{i\theta/2}$, so that $x \sim |s_1|^2 e^{i\theta}$ when $|s_1| \to \infty$, we find

$$\frac{1}{4} - \frac{27a^2 \sin^2 \frac{\theta}{2}}{16|x|} + O\left(\frac{1}{|x|^{3/2}}\right) \leq \left| \frac{\mathcal{M}(\tilde{z}, a) - \alpha_0}{\alpha_1(a)a^2} \right| \leq \frac{|x|}{27a^2 \sin^2 \frac{\theta}{2}} - \frac{1}{4} \cot^2 \theta + O\left(\frac{1}{|x|^{1/2}}\right). \tag{6.13}$$

Let us comment on this form. First, since $|x| \sim |s_1|^2$, the upper bound for fixed $\theta$ is the $|s_1|^2$ bound

on the amplitude. Next, note the the important $\sin^2 \frac{\theta}{2}$ factor. If we took $s_1$ to be real and positive, then the upper bound would be trivial. However, the real $s_1$-axis gets mapped to boundary of the unit disc, which is not a part of the open disc. Thus to use these bounds, we have to keep $\theta \in (0, 2\pi)$, with the end-points excluded. Next, note the more interesting lower bound which begins with a constant! Finally, and importantly, the bound involves $\alpha_1(a)a^2$. In terms of Wilson coefficients, this involves only $\mathcal{W}_{01}, \mathcal{W}_{10}$. Thus, in the Regge limit, we have the following interesting bound

$$\frac{27a^2}{4} |a\mathcal{W}_{01} + \mathcal{W}_{10}| \lesssim |\mathcal{M}(\tilde{z}, a) - \mathcal{M}(0, a)| \lesssim \frac{|s_1|^2}{\sin^2 \frac{\theta}{2}} |a\mathcal{W}_{01} + \mathcal{W}_{10}| . \tag{6.14}$$

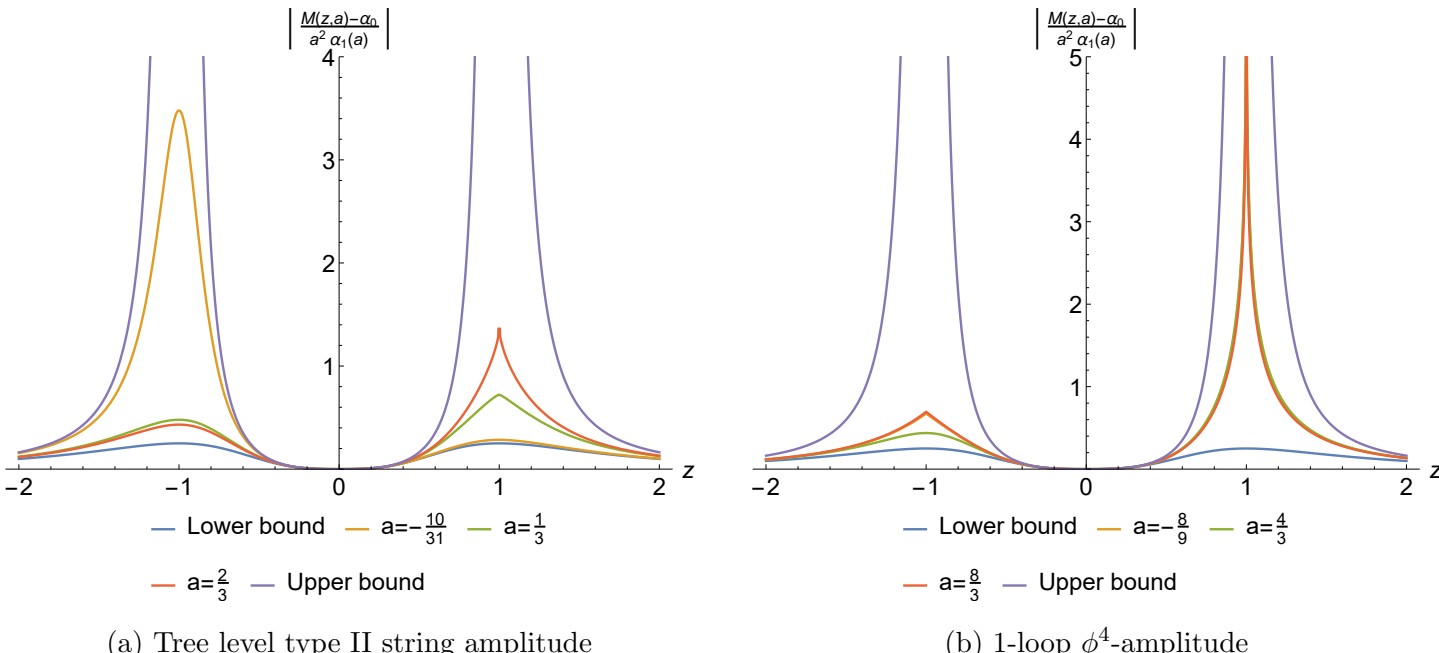

(a) Tree level type II string amplitude        (b) 1-loop $\phi^4$-amplitude

Figure 5: Bounds on amplitude, as in theorem (6.1), are satisfied by Tree level type II string amplitude and 1-loop $\phi^4$-amplitude.

# 7 Univalence of the EFT expansion

So far, on the QFT side, we have focused on results motivated by the Bieberbach conjecture but which one can derive by using the crossing symmetric dispersion relation. All the QFT conditions we have derived so far hint at univalence. Establishing univalence in generality is a hard question and beyond the scope of our present work. Nevertheless, we can investigate scenarios where univalence is guaranteed to not hold. We will begin our investigations using Szegö's theorem[14]. For definiteness, consider the low energy expansion of the 2-2 dilaton scattering in type II string theory with the massless pole subtracted. We will consider

$$f(\tilde{z}, a) = \frac{M(\tilde{z}, a) - M(0, a)}{\partial_{\tilde{z}} M(\tilde{z}, a)|_{\tilde{z}=0}} , \tag{7.1}$$

expanded around $\tilde{z} = 0$. This corresponds to a low energy expansion of the amplitude since $\tilde{z} \sim 0$ corresponds to $x \sim 0, y \sim 0$. If $M(\tilde{z}, a)$ was univalent inside a disc $D$ of radius $R$, for a certain range of $a$, then $f(\tilde{z}, a)$ should be locally univalent inside $D$. This means that the absolute value of

---

[14]We thank P. Raman for numerous suggestions and discussions pertaining to this section.

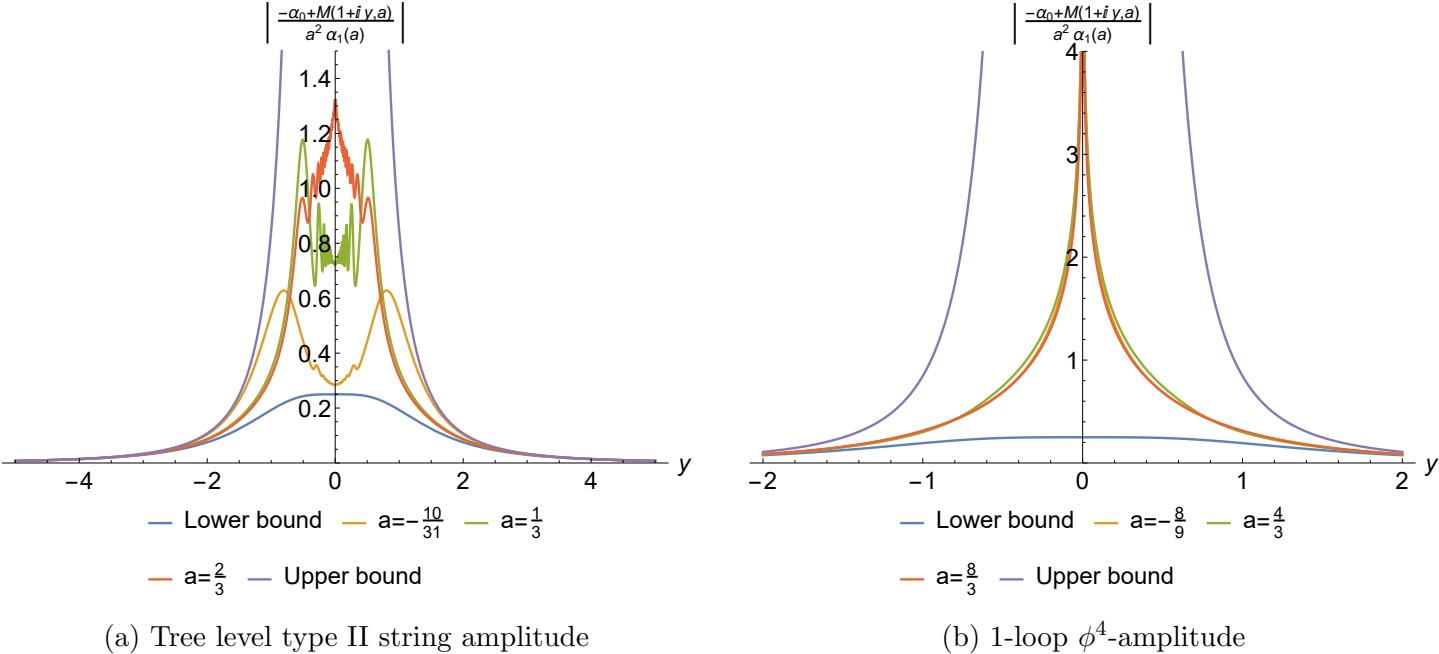

(a) Tree level type II string amplitude

(b) 1-loop $\phi^4$-amplitude

Figure 6: Bounds on amplitude, as in theorem (6.1), are satisfied by Tree level type II string amplitude and 1-loop $\phi^4$-amplitude. These bounds on amplitude valid for complex $\tilde{z}$.

the smallest root ($\tilde{z} = \zeta_{min}$) of $\partial_{\tilde{z}} f(\tilde{z}, a) = 0$ should be greater than $1/4$ to satisfy Szegö's theorem. This can be easily checked using Mathematica to some high power in the expansion in $\tilde{z}$. The plot in fig.(7a) shows that univalence of the full amplitude is only possible in the range $a \in (-1/3, 2/3)$ as one may have anticipated from our previous discussion. In fig.(7b) we show monotonicity of $\zeta_{min}$ as a function of $n$ for small $a$. This is intuitive in the sense that for higher $n$'s, we are putting in more number of terms in the EFT expansion, as a result of which the radius of the disk, within which there is potential univalence, increases. We should point out, however, that for slightly larger values of $a$, for instance, $a > 0.05$, there are other features that arise in the plot, which do not respect monotonicity—the physical implication of this finding is unclear to us. For 1-loop $\phi^4$, our findings are qualitatively similar. In appendix D, we comment on the Nehari conditions.

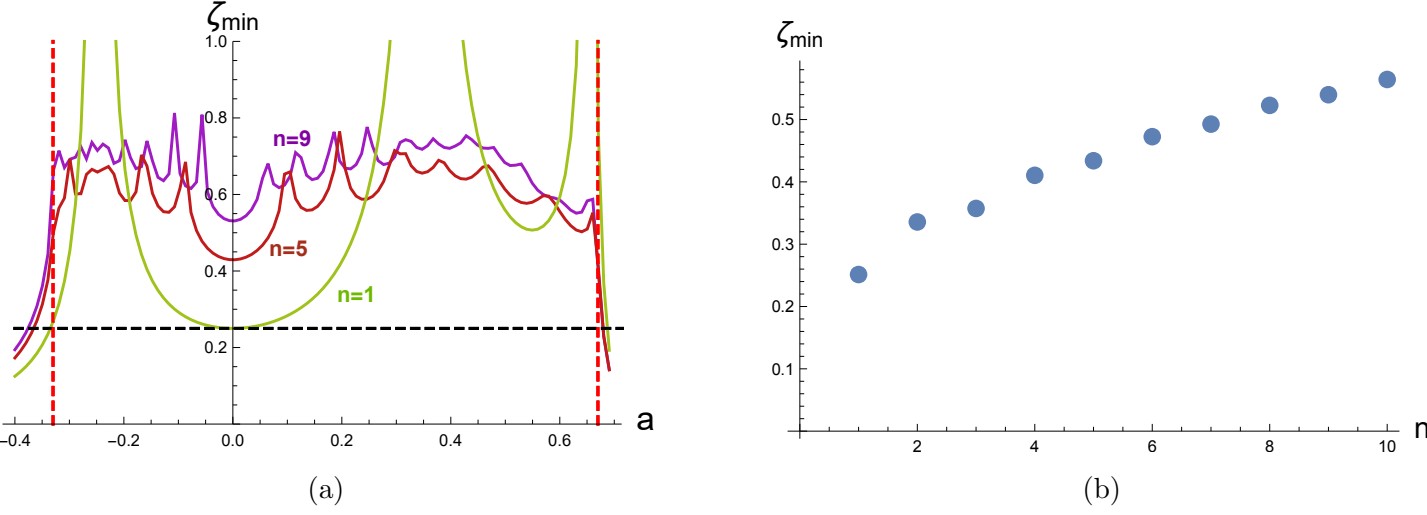

(a)

(b)

Figure 7: Testing univalence using Szegö's theorem. (a) Plot of $\zeta_{min}$ vs $a$. This demonstrates that for the string amplitude to be potentially univalent, $-1/3 < a < 2/3$ must hold. (b) Plot of $\zeta_{min}$ vs $n$ for $a = 0.02$.

## 7.1 Grunsky inequalities and EFT expansion

From the expansion in (3.2), one can relate the $\mathcal{W}_{p,q}$ with the $\omega_{j,k}$ in (2.9). One can easily check that for each $N$ in (2.9), $p + q \leq 2N + 1$ numbers of $W_{p,q}$ appears in (2.9). Therefore, in order to hold the (2.9) till $N$, for (6.1), one has to retain terms in EFT expansion (3.2) up to

$$p + q \geq 2N + 1. \tag{7.2}$$

Before delving into generalities, we begin by considering a toy problem.

### 7.1.1 A toy example: scalar EFT approximation

Since the general case of the univalence of the convex sum of univalent functions is not entirely clear (or more appropriately known to us), let us consider a toy problem below. This problem is enlightening for several reasons. For starters, the amplitudes we consider below are in two "standard" forms. Both of these were considered in [10] to study scalar EFTs, and it was found that scalar EFTs could be approximated as a convex sum of the amplitudes below. Thus consider the sum

$$\mathcal{M}^{(toy)}(s_1, s_2) = \mu_1 \mathcal{M}_1(s_1, s_2) + \mu_2 \mathcal{M}_0(s_1, s_2) \tag{7.3}$$

where

$$
\begin{aligned}
\mathcal{M}_0(s_1, s_2) &= \frac{1}{M_1^2 - s_1} + \frac{1}{M_1^2 - s_2} + \frac{1}{M_1^2 + s_1 + s_2} - \frac{3}{M_1^2} = \frac{27a^2 \tilde{z} \left(2M_1^2 - 3a\right)}{27a^3\tilde{z} - 27a^2 M_1^2 \tilde{z} - M_1^6 \left(\tilde{z} - 1\right)^2} \\
\mathcal{M}_1(s_1, s_2) &= -\frac{1}{\left(M_2^2 - s_1\right)\left(M_2^2 - s_2\right)\left(M_2^2 + s_1 + s_2\right)} = \frac{\left(\tilde{z} - 1\right)^2}{27a^3\tilde{z} - 27a^2 M_2^2 \tilde{z} - M_2^6 \left(\tilde{z} - 1\right)^2}.
\end{aligned}
\tag{7.4}
$$

In the range (we are considering real $a$ only), $-\frac{M_{1,2}^2}{3} < a < \frac{2M_{1,2}^2}{3}$, individually $\mathcal{M}_0(s_1, s_2)$, $\mathcal{M}_1(s_1, s_2)$ do not have any singularity inside the unit disk. A straightforward calculation shows for both of them $\omega_{j,k} = -\frac{\delta_{j,k}}{k}$ for $j > 0, k > 0$. Therefore, $\mathcal{M}_0(s_1, s_2)$, $\mathcal{M}_1(s_1, s_2)$ are individually univalent inside the unit disk when the restriction on $a$ holds. Now the sum of $\mathcal{M}_0(s_1, s_2)$, $\mathcal{M}_1(s_1, s_2)$, we denoted $\mathcal{M}^{(toy)}(s_1, s_2)$ does not have any singularity inside the unit disk. We can check the univalence of the combination using the Grunsky inequalities again. Quite remarkably, if $M_1 = M_2$, we again find that $\omega_{j,k} = -\frac{\delta_{j,k}}{k}$ for $j > 0, k > 0$. Therefore, for any $\lambda_1, \lambda_2$ for the range of $a$ above, the combination is univalent inside the disk! When $M_1 \neq M_2$ we get nontrivial constraints for univalence. Already at $N = 1$, the Grunsky inequality (2.9) leads to

$$\left| 1 - \frac{729a^4 \mu_1 \mu_2 \left(M_2^2 - M_1^2\right)^2 \left(a - M_1^2\right)\left(2M_1^2 - 3a\right)^3}{M_1^{10} \left(\mu_1(a - M_1^2) + \mu_2 M_1^6(2M_1^2 - 3a)\right)^2} \right| \leq 1, \tag{7.5}$$

where we have expanded around $M_1^2 = M_2^2$ and retained only the leading term. This leads to a constraint on $\mu_1, \mu_2$ in terms of $M_1, M_2, a$. One interesting point to make note of is the following: for $a \sim 0$, for the above condition to hold, we will need $\mu_1 \mu_2 < 0$. This is a consequence of unitarity and conforms with the signs in [10]. We leave a detailed investigation of such constraints for the future[15]. Next, we will consider expanding the amplitude around $a \sim 0$ and will find that to leading order the Grunsky inequalities hold.

---

[15]In light of such complications, it seems to us that it will be more useful in physics to think about an approximate notion of univalence, may be saying that a function is approximately univalent if the first few $N$'s in the Grunsky inequalities hold.

### 7.1.2 Proof of univalence of $f(\tilde{z}, a)$ for $|a|$ small

Using the expansion (3.2), one can calculate the Grunsky coefficients, $\{\omega_{j,k}\}$, for $f(\tilde{z}, a)$ in leading order of small $a$ to obtain

$$\omega_{j,k} = -\frac{\delta_{j,k}}{k} + \frac{729a^4 jk \left(\mathcal{W}_{1,0}\mathcal{W}_{3,0} - \mathcal{W}_{2,0}^2\right)}{\mathcal{W}_{1,0}^2} + O(a^5). \tag{7.6}$$

From positivity [9, eq (6.4)], $\mathcal{W}_{1,0}\mathcal{W}_{3,0} \geq \mathcal{W}_{2,0}^2$. Therefore second term in the above equation is positive. Therefore, we get

$$|\omega_{j,k}| \leq \frac{\delta_{j,k}}{k} \text{ , for j=k} . \tag{7.7}$$

The off-diagonal terms $(j \neq k)$ starts at $O(a^4)$. From (2.9), we can say that for small $a$, Grunsky inequalities are satisfied by $f(\tilde{z}, a)$ in (6.1).

We can push the Grunsky inequalities further. If we assume that perturbatively around $a = 0$, univalence should hold, then we can derive nonlinear inequalities by making clever choice for the complex parameters $\lambda_k$ in (2.9). For instance, for $N = 2$ in (2.9), choosing $\lambda_2 = -\lambda_1/2$ and $\lambda_1, \lambda_2$ as real, we easily find the following complicated nonlinear inequality

$$-3\mathcal{W}_{2,0}^4 + 8\mathcal{W}_{1,0}\mathcal{W}_{2,0}^2\mathcal{W}_{3,0} - 4\mathcal{W}_{1,0}^2\mathcal{W}_{2,0}\mathcal{W}_{4,0} - 3\mathcal{W}_{1,0}^2\mathcal{W}_{3,0}^2 + 2\mathcal{W}_{1,0}^3\mathcal{W}_{5,0} \geq 0 . \tag{7.8}$$

We have verified that the string amplitude, as well as the pion S-matrices, satisfy this inequality.

## 7.2 Bounds in case of EFTs:

In an EFT, usually, the Lagrangian is known up to some energy scale. From that information, one can calculate the amplitude up to that scale. In such cases, we can subtract off the known part of the amplitude. These steps result in a shift in the lower limit of the dispersion integral (3.3) by the scale $\delta_0$, namely $\mu \to \mu + 3\delta_0/2$ (see [4]). Therefore, making this replacement in (5.1), we have

$$-\frac{9}{4\mu + 6\delta_0} < \frac{\mathcal{W}_{0,1}}{\mathcal{W}_{1,0}} < \frac{9}{2\mu + 3\delta_0} . \tag{7.9}$$

Here $\mathcal{W}$ are the Wilson coefficients of the amplitude with subtractions. Now notice that if we consider $\delta_0 \gg \mu$ then we have

$$-\frac{3}{2\delta_0} < \frac{\mathcal{W}_{0,1}}{\mathcal{W}_{1,0}} < \frac{3}{\delta_0} . \tag{7.10}$$

Let us compare this to [10]. Converting their results to our conventions, we find that the lower bound above is identical to their findings–this is corroborated by the results of [9] as well as what arises from crossing symmetric dispersion relations [4]. The other side of the bound is more interesting. The strongest result in $d = 4$ in [10] places the upper bound at $\approx 5.35/\delta_0$ in our conventions. Their approach also makes the bound spacetime dimension dependent. Now remarkably, the bound we quote above and the $d \gg 1$ limit of [10] are identical! In EFT approaches, one takes the so-called null constraints or locality constraints and expands in the limit $\delta_0 \gg \mu$. It is possible that a more exact approach building on [10] will lead to a stronger bound as in (7.10).

Let us now comment on the behaviour in the figure (3). First, notice that all S-matrices appear to respect the upper bound we have found above; for the S-matrix bootstrap results, we set $\delta_0 = 0$.

For comparison, note that for 1-loop $\phi^4$, and 2-loop chiral perturbation theory, we have

$$\left(\frac{\mathcal{W}_{0,1}}{\mathcal{W}_{1,0}}\right)_{\phi^4} \approx -0.315\,, \quad \left(\frac{\mathcal{W}_{0,1}}{\mathcal{W}_{1,0}}\right)_{\chi-PT} \approx -0.135\,, \tag{7.11}$$

both in units where $m = 1$. These numbers would be closer to the lower black dashed line in the figure (3), which is the bound that is common in all approaches so far. The upper black dashed line is what we find in the current paper. For future work, it will be interesting to search for an interpolating bound as in (7.9) which enables us to interpolate between $\delta_0 = 0$ and $\delta_0 \gg \mu$.

# 8   Discussion

In this paper, we have examined a potentially remarkable correspondence between aspects of geometric function theory and quantum field theory. We believe we have just scratched the surface. There are many interesting questions to pursue in the near future. By no stretch of the imagination is our examination of the vast mathematics literature on univalent functions exhaustive. While we believe we have identified some of the interesting mathematics theorems which have either a QFT counterpart or applications in QFT, there must be a great many connections waiting to be discovered. Let us first recapitulate what we accomplished in this paper:

- We found QFT counterparts to (1.2) and (1.3). In QFT, we made use of the crossing symmetric dispersion relation and unitarity. The only place we used univalence was for the kernel in the crossing symmetric dispersion relation, which we proved was univalent inside the open disk $\mathbb{D}$ for a range of $a$. In deriving the appropriate kernel for the dispersion relation, we assumed that $\mathcal{M}(s,t)$ in the Regge limit went like $o(s^2)$. However, the upper bound in (1.3) is stronger since it applies not just in this limit. To see this, recall that the Regge limit was $\tilde{z} \to 1$, and the bound applies more generally.

- The univalence of the kernel enabled us to derive upper bounds on the Taylor coefficients of the scattering amplitude via application of the de Branges's theorem (Bieberbach conjecture) to the kernel. These bounds can be used to obtain inequalities concerning the Wilson coefficients, and we found strong bounds which are respected by all theories considered in this paper, which include 1-loop $\phi^4$, pion S-matrix bootstrap (which included a plethora of examples which respects unitarity, crossing symmetry and has information about the standard model $\rho$-meson mass) and even the massless pole subtracted string tree-level dilaton scattering.

- We further derived two-sided bounds on the scattering amplitude. In deriving the upper bound, we used the univalence of the kernel in the form of the Koebe growth theorem. The upper bound, expressed in terms of the usual Mandelstam variables, translates to, for large $|s|$, fixed $t$, $|\mathcal{M}(s,t)| \lesssim s^2$.

- We proved that to leading order in $a$, around $a \sim 0$ the scattering amplitude is univalent as it respects the Grunsky inequalities.

Here is our immediate wish-list:

- The crossing symmetric kernel was univalent for regions of complex $a$. We did not examine this in detail in this paper since we wanted to make use of the positivity of the absorptive part. It will be important to ask if the univalence of the full amplitude holds for complex $a$.

- In examples that do not have three channel crossing symmetry, like open string scattering amplitudes for Yang-Mills, or even Moller scattering, it will be interesting to identify the appropriate variable in which univalence can be studied.

- In light of our proof of univalence to leading order in $a$, a more detailed study of what is known about the convex sum of univalent functions should be made.

- While the power of univalence cannot be denied, in the mathematics literature, there are also interesting and potentially powerful theorems (and conjectures) corresponding to multivalent functions. The connection between multivalence and QFT should also be examined.

As physicists, we should ask if univalence is a new/stronger condition or if it is possible to simply prove this holds (for a range of $a$ values) using standard QFT dispersion techniques. While we do not have a conclusive opinion about this, we should point out that (7.5) holds near $a \sim 0$ only for unitary theories. Further, we have implicitly used unitarity to derive our QFT results since it entered in showing positivity of the absorptive part. So maybe we can sharpen this question by asking: Does dispersion relation and univalence imply unitarity?

It will be also interesting to expand the S-matrix bootstrap numerics to more general cases (for instance, in higher dimensions, varying the $\rho$-meson mass etc) to examine the inequalities considered in this paper[16]. In a related vein, the CFT Mellin amplitudes [24] also admit a crossing symmetric representation [5]. One can ask related questions in these cases as well. In [25] CFT correlators in position space in the diagonal limit were shown to be two-sided bounded, reminiscent of (1.3). It is tempting to think that there will be a similar story at work there.

It took around 70 years for the (now more than 100 year old) Bieberbach conjecture to be proved in generality. We found an interesting connection with physics; it is highly likely that there are more gems and hidden treasures buried waiting to be discovered!

# Acknowledgments

We thank Faizan Bhat, Rajesh Gopakumar, Miguel Paulos, Prashanth Raman and Sasha Zhiboedov for useful discussions. We thank Yang-Hui He and Rajesh Gopakumar for comments on the draft. We especially thank Shaswat Tiwari for helping us with the S-matrix bootstrap numerical checks for the Wilson coefficients. We acknowledge partial support from a SPARC grant P315 from MHRD, Govt of India.

# A    Univalence in Physics

As conveyed earlier, the discussion of univalence in physics is scarce to find in the literature. However, such endeavours have been up taken in the distant past and in some recent work. In this appendix, we will give concise reviews of such papers [6, 7, 8].

## Univalence in forward scattering

To the best of our knowledge, univalence in the context of high energy scattering amplitude was first considered in the mid-1960s in the background of axiomatic field theory considerations. Khuri

---

[16] The Grunsky inequalities superficially have some similarities with the non-linear inequalities arising in the EFT-hedron story in [23].

and Kinoshita [6] constructed a univalent function out of the *forward scattering amplitude*. We will summarize their analysis in what follows. Starting from the forward scattering amplitude $\mathcal{M}(s,0)$, one can construct the function

$$g(s) = \int_0^s ds' \, \frac{\mathcal{M}(s',0) - \mathcal{M}(0,0)}{(s')^2}, \tag{A.1}$$

which can be *proved to be univalent in the upper-half $s$ plane, i.e. for $\Im(s) > 0$*. See [6] for the detailed argument of the proof.

Univalent functions satisfy sharp inequalities. However, usually, these inequalities are stated for univalent functions on the open unit disc. But, by Riemann mapping theorem, every simply connected, proper, open subset of the complex plane can be bi-holomorphically (holomorphic bijective mapping) mapped onto the open disc $\mathbb{D}$. Thus, we can map the upper half-$s$-plane bi-holomorphically to $\mathbb{D}$. One such bi-holomorphic mapping is

$$w(s) := \frac{s - i\lambda}{s + i\lambda}, \quad \lambda > 0. \tag{A.2}$$

Under this map, the upper-half $s$ plane is mapped to the unit disc $|w| < 1$, in the $w$ plane with $s = i\lambda$ mapped to the origin $w = 0$. It is to be emphasized that the mapping is defined *for fixed $\lambda$*[17]. Now one can consider the function $\varphi(w)$ defined by

$$\varphi(w) := \frac{g(s(w)) - g(i\lambda)}{2i\lambda \, g'(i\lambda)}. \tag{A.3}$$

Because $g(s)$ is univalent in the upper half-$s$ plane, $g'(i\lambda) \neq 0$ necessarily. Thus $\varphi(w)$ is well-defined over $|w| < 1$. Further, since $w(s)$ is a bi-holomorphism, $\varphi(w)$ is a univalent mapping on the open disc $|w| < 1$. Also note that, $\varphi(0) = 0$ and $\varphi'(0) = 1$. Thus, $\varphi(w)$ admits a power series representation of the form

$$\varphi(w) = w + \sum_{k=2}^{\infty} \gamma_k \, w^k. \tag{A.4}$$

Thus, it is a schlicht function, $\varphi \in \mathcal{S}$. And now one can apply Koebe growth theorem to this function to estimate bounds on $g(s)$.

## Univalence in flux-tube bootstrap

Recently in [7], possible role of univalent functions has been explored in the context of flux-tube S-matrix bootstrap. We would like to thank Andrea Guerrieri for drawing our attention to this work. We briefly summarize the investigation as below.

Excitations of the flux-tube can be modelled by massless particles called branon. Their scattering is described by 2D massless S-matrix [18]

$$S(s) = e^{2i\delta(s)}, \tag{A.5}$$

with

$$2\delta(s) = \frac{s}{4} + \gamma_3 s^3 + \gamma_5 s^5 + \gamma_7 s^7 + i\gamma_8 s^8 + O(s^9). \tag{A.6}$$

Here $\gamma_3, \gamma_5, \gamma_7$ are non-universal parameters which parametrize the theory space. On the other hand

---

[17]In fact, this is the *unique* mapping with the property $w(i\lambda) = 0$ and $\mathrm{Arg}.[w'(i\lambda)] = 3\pi/2$ for *fixed $\lambda$*.
[18]Note that in 2D, we have a single Mandelstam variable $s$.

$\gamma_8 \propto \gamma_3^3$ is not independent, see [7] for details.

The S-matrix $S(z)$ is a holomorphic function from the upper half-plane $\mathbb{H}^+ := \{z|\ Im(z) > 0\}$ to the unit disk $\mathbb{D}$. This can be seen as following. From unitarity, $|S(z)| \leq 1$ for $z \in \mathbb{R}$. Then applying maximum modulus principle, we have

$$S(z) \leq 1,\ z \in \mathbb{H}^+ \tag{A.7}$$

The holomorphic map $S : \mathbb{H}^+ \to \mathbb{D}$ satisfies the Schwarz-Pick inequality:

$$\left| \frac{S(z_1) - S(z_2)}{1 - S(z_1)\overline{S(z_2)}} \right| \leq \left| \frac{z_1 - z_2}{z_1 - \bar{z}_2} \right|,\ z_1, z_2 \in \mathbb{H}^+ \tag{A.8}$$

This can be obtained by applying Schwarz-Pick lemma to the holomorphic map $S \circ W^{-1} : \mathbb{D} \to \mathbb{D}$, where $W : \mathbb{H}^+ \to \mathbb{D}$ is Cayley transform defined by

$$W(z) = \frac{z - i}{z + i}, z \in \mathbb{H}^+ \tag{A.9}$$

The equality in (A.8) above is satisfied if and only if $S$ is a holomorphic isomorphism or equivalently an univalent function.

We can now consider the function

$$S^1(z|w) := \left[ \frac{S(z) - S(w)}{1 - S(z)\overline{S(w)}} \right] \left[ \frac{z - w}{z - \bar{w}} \right]^{-1} \tag{A.10}$$

Then, the Schwarz-Pick inequality (A.8) gives

$$\left| S^1(z|w) \right| \leq 1\, , \forall z, w \in \mathbb{H}^+ \tag{A.11}$$

Now inserting (A.5) and (A.6) into (A.11) above and expanding for small imaginary $z$ and $w$, we get

$$S^1(ix|iy) = -1 + \left( \frac{1}{96} + 8\gamma_3 \right) xy + \cdots \geq -1 \tag{A.12}$$

This leads to the bound

$$\gamma_3 \geq -\frac{1}{768} \tag{A.13}$$

The main important point in connection to univalence is that the *bound is saturated by univalent S-matrix*. These functions have been called single CDD-zero functions in [7].

## Hydrodynamical bounds from univalence

In the recent work [8], the theory of univalent functions was put to use for deriving bounds on the hydrodynamic transport coefficients. The starting point is the frequency-momentum dispersion relation, $\omega(\mathbf{q}^2)$, obtained from linearised hydrodynamics. Here $\omega$ is the frequency, and $\mathbf{q}^2$ is the momentum squared of a collective mode: diffusion or sound. In a hydrodynamical theory preserving

spatial rotations, the classical[19] $\omega(\mathbf{q}^2)$ are given by infinite series of the form

$$\omega_{\text{diff}}\left(z \equiv \mathbf{q}^2\right) = -i \sum_{n=1}^{\infty} c_n z^n = \frac{f_{\text{diff}}(z)}{i},$$

$$\omega_{\text{sound}}^{\pm}\left(z \equiv \sqrt{\mathbf{q}^2}\right) = -i \sum_{n=1}^{\infty} a_n e^{\pm \frac{i\pi n}{2}} z^n = f_{\text{sound}}^{\pm}(z), \tag{A.14}$$

with $a_n$, $c_n \in \mathbb{R}$ for all $n \geq 1$. The coefficient $c_1 = D$ is the diffusivity, and $a_1 = v_s$ is the speed of sound. Treating $z$ as a complex variable in both of the above equations, one investigates the domains in the complex $z$ plane on which $\omega_{\text{diff}}(z)$ and $\omega_{\text{sound}}^{\pm}(z)$ are univalent. The main tool to establish this is to use $\text{Re} f'(z) > 0$ which is a sufficient condition for an analytic function to be univalent, which translates into conditions on the group velocity. Explicit checks for situations where such conditions holds were done using holography–for more details see [8]. Then, in that domain one can use univalence to write bounds via Koebe growth theorem:

$$\frac{|\omega_0|\left(1 - |\zeta_0|\right)^2}{|\zeta_0|\,|\partial_\zeta \varphi^{-1}(0)|} \leq (D \text{ or } v_s) \leq \frac{|\omega_0|\left(1 + |\zeta_0|\right)^2}{|\zeta_0|\,|\partial_\zeta \varphi^{-1}(0)|}, \tag{A.15}$$

where $\zeta := \varphi(z)$ is the conformal mapping from domain of univalence to open unit disk, and $\varphi(0) = 0$. $z = z_0$ is a point in the domain of univalence such that $\omega_0 := \omega(z_0)$ is known and $\zeta_0 = \varphi(z_0)$. Also, one can write bounds from de Branges theorem

$$\left| c_2 + \frac{D}{2} \frac{\partial_\zeta^2 \varphi^{-1}(0)}{[\partial_\zeta \varphi^{-1}(0)]^2} \right| \leq \frac{2D}{|\partial_\zeta \varphi^{-1}(0)|}. \tag{A.16}$$

Using the relation between transport and chaos, which has been established in large-$N$ theories, namely via pole skipping considerations, which relate frequency at a specific complex momentum with the Lyapunov exponent, interesting bounds were derived in [8] which give two-sided bounds on diffusivity in terms of the Lyapunov exponent. The main challenge in [8] as well as in the present paper is to identify conditions where univalence holds. As pointed out in the main text, we were able to get mileage by knowing where the kernel appearing in the dispersion relation was univalent. Further constraints will arise on establishing the precise conditions where the full amplitude (which is a convex sum of univalent functions) is univalent.

# B    Various Amplitudes

## B.1    Tree level type II superstring theory amplitude

The low energy expansion of the type II superstring amplitude is well known, see for example [26] for a recent discussion. The amplitude after stripping off a kinematic factor and subtracting off the massless pole is given below. This is what we will use. In order to facilitate expansion, it is also useful to recast the Gamma function in terms of an exponential of sum of Zeta functions as in [26].

$$\mathcal{M}^{(cl)}(s_1, s_2) = -\frac{\Gamma(1 - s_1)\,\Gamma(1 - s_2)\,\Gamma(s_1 + s_2 + 1)}{s_1 s_2 (s_1 + s_2)\,\Gamma(s_1 + 1)\,\Gamma(-s_1 - s_2 + 1)\,\Gamma(s_2 + 1)} + \frac{1}{s_1 s_2 (s_1 + s_2)} \tag{B.1}$$

---

[19]The classical theory is devoid of any stochastic noise or loop corrections which lead to breakdown of analyticity.

| $\mathcal{W}_{p,q}$ | q=0 | q=1 | q=2 | q=3 | q=4 | q=5 |
|---|---|---|---|---|---|---|
| p=0 | 2.40411 | -2.88988 | 2.98387 | -2.99786 | 2.99973 | -2.99997 |
| p=1 | 2.07386 | -4.98578 | 7.99419 | -10.9987 | 13.9998 | -17. |
| p=2 | 2.0167 | -6.99881 | 14.9984 | -25.9995 | 39.9999 | -57. |
| p=3 | 2.00402 | -9.00023 | 23.9996 | -49.9998 | 89.9999 | -147. |
| p=4 | 2.00099 | -11.0002 | 34.9999 | -84.9999 | 175. | -322. |
| p=5 | 2.00025 | -13.0001 | 48. | -133. | 308. | -630. |

Table 1: $\mathcal{W}_{p,q}$ for tree level type II superstring theory amplitude

Note that we have stripped off the kinematic factor $x^2 = (s_1 s_2 + s_2 s_3 + s_1 s_3)^2$. Had we retained it then the graviton pole subtracted amplitude in the Regge limit would have behaved like $|s_1|^2/t$ so that the dispersion relation would need three subtractions. Therefore, it is important that we remove this kinematic factor in what we do. The $a_\ell$'s with this factor removed continue to be positive–which is the main thing we used in our derivation.

## B.2    1-loop $\phi^4$ amplitude

We just note the well-known standard result for the 1-loop $\phi^4$ amplitude.

$$\mathcal{M}^{(\phi^4)}(s_1, s_2) = -\frac{2\sqrt{s_1 - \frac{8}{3}}\tanh^{-1}\left(\frac{\sqrt{s_1 + \frac{4}{3}}}{\sqrt{s_1 - \frac{8}{3}}}\right)}{\sqrt{s_1 + \frac{4}{3}}} - \frac{2\sqrt{s_2 - \frac{8}{3}}\tanh^{-1}\left(\frac{\sqrt{s_2 + \frac{4}{3}}}{\sqrt{s_2 - \frac{8}{3}}}\right)}{\sqrt{s_2 + \frac{4}{3}}} - \frac{2\sqrt{s_3 - \frac{8}{3}}\tanh^{-1}\left(\frac{\sqrt{s_3 + \frac{4}{3}}}{\sqrt{s_3 - \frac{8}{3}}}\right)}{\sqrt{s_3 + \frac{4}{3}}}$$

(B.2)

| $\mathcal{W}_{p,q}$ | q=0 | q=1 | q=2 | q=3 | q=4 | q=5 |
|---|---|---|---|---|---|---|
| p=0 | -5.22252 | -0.0209238 | 0.000401094 | -0.0000116118 | 3.9934× $10^{-7}$ | -1.5104× $10^{-8}$ |
| p=1 | 0.0663542 | -0.0023309 | 0.0000983248 | -4.4442× $10^{-6}$ | 2.0832× $10^{-7}$ | - |
| p=2 | 0.00344623 | -0.00027954 | 0.00001862 | -1.1521× $10^{-6}$ | - | - |
| p=3 | 0.000267396 | -0.0000348355 | 3.1948× $10^{-6}$ | -2.5174× $10^{-7}$ | - | - |
| p=4 | 0.0000245812 | -4.4442× $10^{-6}$ | 5.2081× $10^{-7}$ | - | - | - |
| p=5 | 2.4827× $10^{-6}$ | -5.7605× $10^{-7}$ | - | - | - | - |

Table 2: $\mathcal{W}_{p,q}$ for 1-loop $\phi^4$ amplitude

## B.3    Amplitude for pion scattering from S-matrix bootstrap

The S-matrix bootstrap puts constraints on pion scattering using unitarity and crossing symmetry. Some additional phenomenological inputs like $\rho$-meson mass or certain theoretical constraints like S/D wave scattering length inequalities are used. For more details, the reader is referred to [20, 21, 22]. The allowed S-matrices are displayed as regions on the Adler-zeros $(s_0, s_2)$ plane. In [21], a river like region of S-matrices on this plane was identified. The chiral perturbation theory appeared to lie close to a kink-like feature near $s_0 = 0.35$. As such this particular S-matrix is of interest to us. In the main text, we have considered a plethora of S-matrices like the lake in [20], the upper and lower boundaries of the river in [21] as well as the more interesting line of minimization (where the total scattering cross-section is minimized for a given $s_0$) in [22]. The amplitude for pion scattering from S-matrix

bootstrap with $s_0 = 0.35$ is given[20] below for $a = 1/2$

$$\mathcal{M}(\tilde{z}, a) = -1.90562 - 55.586\tilde{z} - 75.7314\tilde{z}^2 - 49.2812\tilde{z}^3 + 3.43872\tilde{z}^4 + 45.4445\tilde{z}^5 + O\left(\tilde{z}^6\right) \quad \text{(B.3)}$$

In table (3), we have listed various $\mathcal{W}_{p,q}$.

| $\mathcal{W}_{p,q}$ | q=0 | q=1 | q=2 | q=3 | q=4 | q=5 |
|---|---|---|---|---|---|---|
| p=0 | -1.90562 | 5.02671 | -0.249527 | 0.0118008 | -0.000555517 | 0.0000262344 |
| p=1 | 5.72161 | 0.395863 | -0.0520982 | 0.00402939 | -0.000264317 | - |
| p=2 | 0.642298 | 0.0217519 | -0.00787377 | 0.000904172 | - | - |
| p=3 | 0.0796397 | -0.000836409 | -0.000995454 | 0.000166504 | - | - |
| p=4 | 0.0101505 | -0.000579411 | -0.000103708 | - | - | - |
| p=5 | 0.0013093 | -0.000136893 | - | - | - | - |

Table 3: $\mathcal{W}_{p,q}$ for pion scattering from S-matrix bootstrap with $s_0 = 0.35$

# C   Grunsky inequalities (2.9) and $s_0 = 0.35$ pion amplitude

Using the table (3), one can check the Grunsky inequalities (2.9) for $N = 2$, with some random $\lambda_1, \lambda_2$. Since we are truncating the sum over Wilson coefficient expansion, if this truncated sum comes from a univalent function (in the range of $-\frac{2\mu}{9} < a < \frac{4\mu}{9}$) and the truncated sum is itself univalent. Therefore, it may be expected that the radius of the disc where univalence holds should be smaller. This translates into the range of $a$, which should be now $-\frac{\mu}{9} < a < \frac{2\mu}{9}$ (or maybe a smaller range of $a$). This can be realized from Szegö's theorem, since $a^{2n}$ always comes with $\tilde{z}^n$, reducing the radius to $1/4$ means reducing the range of $a$ by $1/2$ for unit disk in $\tilde{z}$-plane. One can see in figure (8) that our expectation matches[21] exactly. The main point of the above discussion is that, *there exists a finite range of $a$ for which the $f(\tilde{z}, a)$ is univalent.*

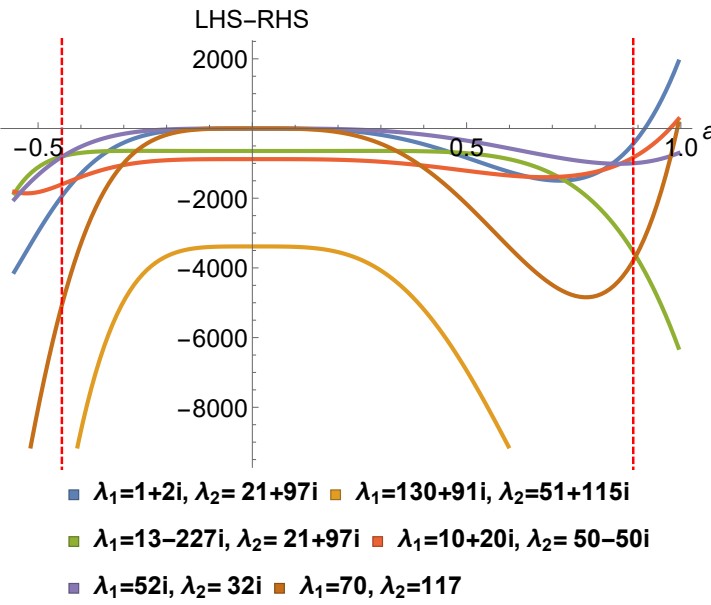

Figure 8: Red lines are the $a = -4/9, 8/9$

---

[20]One can write these kind of expansion for a general $a$ upto desired order in $\tilde{z}$. To minimize numerical errors in our calculations, we had to rationalize upto 20 decimal place.

[21]There can be some random $\lambda_1, \lambda_2$ for which the curves can be slightly below the line $a = 8/9$

# D  Constraints from (5.3)

Suppose we consider $\mathcal{W}_{0,1}, \mathcal{W}_{1,1}, \mathcal{W}_{1,0}, \mathcal{W}_{2,1}, \mathcal{W}_{1,2}\mathcal{W}_{2,0}, \mathcal{W}_{3,0}$ as given. We can constrain $\mathcal{W}_{0,3}$, using (5.3). See figure (9).

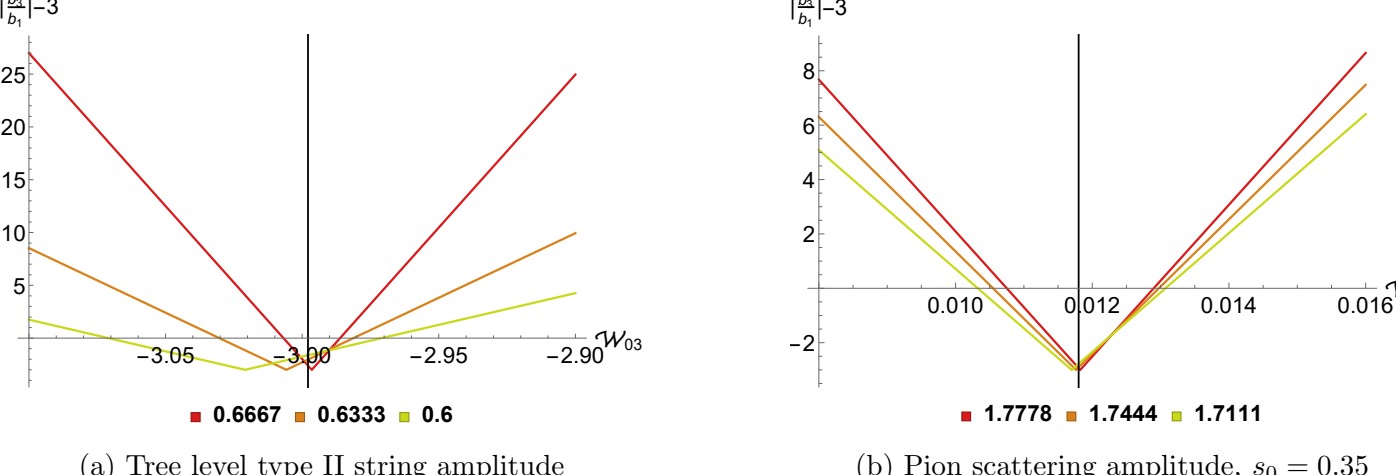

(a) Tree level type II string amplitude          (b) Pion scattering amplitude, $s_0 = 0.35$

Figure 9: Constraints on Wilson coefficient $W_{0,3}$ using (5.3), where $\mathcal{W}_{0,1}, \mathcal{W}_{1,1}, \mathcal{W}_{1,0}, \mathcal{W}_{2,1}, \mathcal{W}_{1,2}\mathcal{W}_{2,0}, \mathcal{W}_{3,0}$ are given. Figure shows that bound on the $\mathcal{W}_{0,3}$. Since $\left|\frac{b_3}{b_1}\right| - 3$ should be less than zero, $\mathcal{W}_{0,3}$ must lie inside the triangle. Black dashed line is the exact answer. Different values of $a$ are indicated with different colours.

# E  Nehari conditions in 1-loop $\phi^4$-theory.

Using the 1-loop $\phi^4$-amplitude, we can check the Nehari conditions. For the range $-4/9 < a < 16/9$, we find that Nehari necessary condition (2.3) always holds. Further, we find that Nehari sufficient condition (2.2) does not always hold within the unit circle. Nevertheless, there are regions where 1-loop $\phi^4$-amplitude respects Nehari sufficient condition (2.2). For example within the radius[22] of $\frac{2}{3}$ for the range $-4/9 < a < 16/9$, the Nehari sufficient condition (2.2) holds.

We can also check the Nehari conditions in $a \sim 0$ region. We can expand the amplitude around $a = 0$, then calculate the Schwarzian derivative (2.14). For example upto $a^4$, we find

$$\{f(z), z\} = -\frac{6}{(z^2 - 1)^2}\left(1 - \frac{0.971 a^4 (z + 1)^4}{(z - 1)^4}\right) \tag{E.1}$$

Of course, for the full range of $a$, the above (E.1) need not to satisfy the Nehari necessary condition (2.3), since this is an EFT type expansion, and by Szegö theorem, we don't expect the univalent to hold in the same range of $-4/9 < a < 16/9$. Further, there always exists a smaller range of $a$, where it is univalent. For example if we consider the radius $1/2$, the above (E.1) satisfies Nehari necessary condition (2.3) for $-0.301 < a < 0.301$. Qualitatively similar features hold for the string amplitude as well.

---

[22]These can be realized replacing $z \to 2z/3$, and check the conditions for the given ranges of $a$.

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
