# Peer review of "Quantum field theory and the Bieberbach conjecture"

_SciPost Physics_

## Round 3 · Referee Report · Anonymous (Referee 1) · 2021-5-27

Strengths

1- Presents interesting results that shed light on other recent developments in the literature. 2- Results are developed in a clear, well structure way. 3- The authors do well in reviewing the literature in sections 3 and 4. 4- The authors compare their various derived general bounds with related results in the literature as well as with particular known S-matrices. 5-The authors do a good job in summarizing their results and presenting future directions in a clean, concise way.

Weaknesses

1- I am puzzled by the claim in Lemma 3.1 that the absorptive part must be strictly positive in the referred domain. What prevent us from having at some $s_1 \in \left[ \frac{2\pi}{3}, \infty\right)$, $a_\ell(s_1) =0 $, $\forall l$ ? That the absorptive part is non-negative seems to be enough for the purposes of the paper.

2- It would feel more satisfying if the discussion in section 7.1.2 were made a bit more comprehensive and systematic. It is a very interesting proposal, and it felt like it could have been pushed a bit further, even though that is listed as a future direction. It would have been very interesting to see how to properly explore $\lambda$s for $N>1$, as well as a more general and expanded discussion on how known S-matrices satisfy (or not) those constrains.

Report

Inspired by results in geometric function theory, the authors derive a number of analytic bounds on the scattering amplitudes of (mostly) massive particles in higher dimensional QFTs. This is a new complementary approach to the higher dimensional S-matrix Bootstrap program and EFT program, and the results obtained by the authors provide a significant contribution to these fields. It also brings attention to important results in mathematics that could play a key role in future developments in the study of QFTs.

Their results raise several questions and open a number of future directions to be explored, such as "what is the role of univalent scattering amplitudes in QFT?". They derive and present their results in a clear, intelligible way. Their arguments are provided with sufficient detail, making the text instructive to the reader. The work is well motivated in the introduction, which provides a context for the results to be developed. Finally, the results and future directions are well summarized in the conclusions. Therefore, this work easily meets expectations and acceptance criteria for this journal.

Requested changes

If agreed on the first weakness listed, the Lemma 3.1 and related parts of the text should be slightly modified.

---

## Round 3 · Referee Report · Anonymous (Referee 2) · 2021-5-28

Strengths

  1. The paper is written in a very clear, self-contained and accessible fashion. For example section 2 introduces all the necessary results from geometric function theory. Section 3 reviews crossing-symmetric dispersion relations. Appendix A reviews previous appearances of univalence in physics.

  2. It brings to light a new connection between the current studies in physics related to constraints on EFTs derived using dispersion relations and some well-known results in mathematics. This interplay has a potential to open new avenues in the future.

Weaknesses

Weaknesses: 1. Univalence is not very well motivated as a property of scattering amplitudes or some other physical observable. On the other hand, the authors show univalence of the dispersion relations kernel which is a very clear and interesting observation.

  1. I have technical questions to some of the arguments in the paper (see below for the details). In particular, I do not think the bound 5.2 (which is a highlight of the paper) is justified. Relatedly, it is not clear to me to what extent the bounds derived in the paper are optimal and if not how to strengthen them.

Report

The paper brings to light an interesting connection between the rich area of mathematics (geometric function theory) and physics (putting bounds on Wilson coefficients). This connection can be enriching for both fields and deserves further investigation.

The sharp observations made in the paper are: a) the kernel in crossing-symmetric dispersion relations is a univalent function; b) the results about univalent functions can be used to derive bounds on Wilson coefficients in view of a).

On the other hand, from a) it follows that physical observables are naturally *convex sums* of univalent functions and it would be very interesting to understand what are the useful lessons to be learned from that.

I recommend publishing the paper in Sci-Post after the technical questions raised below are addressed. These concern certain technical details in the paper and might affect some of the key results reported in the paper.

In particular, the most important is point 8 below.

Requested changes

My comments will be in the order of their appearance in the draft:

  1. (1.5) change W_1,0 to calligraphic.

  2. top of p.5 it should be "univalent" instead of "equivalent"

  3. footnote 5. it is a bit subtle, "schlicht" is more like "plain"(in the sense of being unsophisticated), rather than "simple"(in the sense of easy).

  4. In equation (2.9) sums start from 1, whereas in (2.6) they start from 0. does it mean that there are no bounds on w_{j,0}?

  5. p.8 I am confused by the words "each and every section of any function." what is meant by this in the context of (2.19)?

  6. In formula (3.15) the argument is based on the fact that for cos theta>1 Gegenbauer polynomials are positive. the argument however requires that the sum over partial waves still converge (which is not always the case for cos theta>1). Could the authors elaborate on this?

  7. In formula 4.12 when applying de Branges' theorem is the bound obtained is optimal? In other words, de Branges' theorem applies to any univalent function, whereas here we know beta_n's explicitly. Can one use this fact to derive a better bound?

8***) I do not understand how 5.2 follows from 5.1 . The authors impose 5.2 as the absence of pole in 5.1. However the pole can be avoided by tuning the numerator so that it has a zero at the same value of a. Therefore I do not think that the authors have derived/justified 5.2 . The same applies to 5.4 and 5.5 and 1.5 quoted in the introduction. This is important since this is one of the new bounds derived in the paper.

  1. figure 4. The black line should be changed to dashed (or the caption should be changed).

  2. figures 5 and 6. It would be helpful if the authors can add more detailed captions (similar to how it is done in the previous figures).

I also have two more general open-ended questions:

  1. Is there anything useful to be learned for physicists from the way de Branges has proved the Bieberbach conjecture?

  2. The mapping from rho to z variable in the conformal bootstrap is given by the Koebe function. Is there any significance in that?

  • validity: good
  • significance: high
  • originality: top
  • clarity: high
  • formatting: good
  • grammar: good

Author:  Parthiv Haldar  on 2021-06-04  [id 1488]

(in reply to Report 3 on 2021-05-28)

  1. Regarding point 7, from our derivation it is not clear if our bounds in eq 4.13 [not eq 4.12] are optimal or not. But some consequences of eq 4.13 leads to bounds on Wilson coefficients stronger than some of the bounds found in [4,9,10]. In ongoing work, we are examining such questions in more detail.

Referred to the open-ended questions, 1. Regarding point 1, we believe the answer is yes. de Branges proof makes use of a Lowener chain of differential equations with a parameter t. We believe that this parameter t is similar to the parameter a that we have in the dispersion relation. The Lowener chain itself is connected with the SLE equations in statistical physics.

  1. Regarding point 2, we have been looking at the CFT problem using Mellin amplitudes and our preliminary findings indicate that the 3d Ising model has a similar story as what we have found in the present paper. We hope to report on the CFT connection some time in the near future.

Since our answer 1 is speculative and 2 is still work in progress, we have refrained from adding these comments to the main draft.

Author:  Parthiv Haldar  on 2021-05-28  [id 1475]

(in reply to Report 3 on 2021-05-28)

Regarding point 8, our conclusions are correct. Let us give a cleaner argument and amplify on the point made just above eq. (5.1).
We have already proved in equation (4.10) that $\alpha_1<0$ for $a\in \left(-\frac{2\mu}{9},0\right) \cup\left(0,\frac{4\mu}{9}\right)$. Now, $\alpha_1=-\mathcal{W}_{10}\left( a \frac{\mathcal{W}_{01}}{\mathcal{W}_{10}}+1\right)$ which follows from eq(3.7). Further, $\mathcal{W}_{10}>0$ which was shown in ref. [4,9,22] of the paper. This immediately implies that $\left(a\frac{\mathcal{W}_{01}}{\mathcal{W}_{10}}+1\right)>0$ for the given range of $a$ quoted above. From this, the bound in eq.(5.2) follows readily. It is to be emphasized that if (5.2) is not satisfied, i.e. if $\frac{\mathcal{W}_{01}}{\mathcal{W}_{10}}$ is outside the range given in (5.2) , then dialing $a$ within the range $-\frac{2\mu}{9}<a<\frac{4\mu}{9},\, a\ne 0$, one can actually make the factor $\left(a\frac{\mathcal{W}_{01}}{\mathcal{W}_{10}}+1\right)$ change sign contradicting $\alpha_1(a)$ being negative in the said range of $a$.

As a sanity check one can use the following lines of code in Mathematica:
FindInstance[-9/16 < w01 < 9/8, w01, 100];
(a w01 + 1) /. %;
Reduce[% > 0 && -8/9 < a < 16/9] or even
FindInstance[-8/9 < a < 16/9, a, 2000];
(a w01 + 1) /. %;
Reduce[% >= 0 , w01] // N

Another nontrivial point is that we did ponder on the argument the referee makes, while writing our paper. What if the denominator had a zero at the same time the numerator had a zero. Suppose the denominator had a zero at $a=a_0$. Then this would mean that the $a_0(a_0 \mathcal{W}_{02}+\mathcal{W}_{11})+\mathcal{W}_{20}=0$ and this would imply a relation between apparently independent Wilson coefficients in quantum field theories. We discounted this possibility as being pathological. Nevertheless, the argument we presented above is a cleaner one and we will be happy to add it to the text.

---

## Round 3 · Referee Report · Anonymous (Referee 3) · 2021-5-29

Strengths

  1. Presents novel insights about the scattering amplitude in QFTs by establishing a connection to univalent functions in mathematics.

  2. Concisely and in a clear manner introduces and reviews mathematical concepts that are not commonly known in the physics community.

  3. The authors check each of the bounds they obtain in multiple examples which is always reassuring.

Weaknesses

Some of the exposition in the paper could be slightly improved to make it easier for the reader (see suggested changes below).

Report

This paper raises the tantalizing prospect that the two to two amplitude $M$ considered as a function of the crossing symmetric variables $(\tilde z,a)$ is univalent in some domain. In other words, in addition to the more extensively studied property of analyticity, it seems that the amplitude may also enjoy the property of injectivity. The authors prove that the kernel in the crossing symmetric dispersion relation is univalent and use this property to show that the amplitude obeys bounds such as the Bieberbach conjecture which are obeyed by univalent functions. More concretely, the authors leverage the above to constrain Wilson coefficients in EFTs.

The paper presents novel ideas and the explanations given are clear and straightforward to follow. I therefore believe it satisfies the criteria to be published in Scipost.

At the same time, I have some suggestions that I hope the authors will take into consideration.

Requested changes

  1. A general comment I would like to make is that the $(z,a)$ coordinates for amplitudes are not very familiar to the community. It would be very helpful for readers to know what the parameter ranges such as $-\frac{2\mu}{9} < a < 0$ correspond to in terms of the more familiar $s$ and $t$. For example if we keep $s \geq \mu$, then the physical s channel scattering range for $t \in [\mu -s, 0]$ corresponds to $a \in (-\infty, -\frac{2\mu}{9}]$.

  2. The range $a \in (-\frac{2\mu}{9}, \frac{2\mu}{3})$ assumed in Lemma 3.1 corresponds to $t \in [0,\mu]$. It would be nice to state that the partial wave expansion of the absorptive part $\mathcal A(s,t)$ converges in this range of values of t (in fact it converges even more than this range, upto the Karplus curve). This would also answer point 6 in referee report 3.

  3. However I believe for the case of $\mu = 0$ the partial wave expansion does not converge for $t >0$. Unless it is assumed like later in the paper that there is a cut off $\delta$ after which the cuts start.

  4. In one of the references https://arxiv.org/abs/2012.04877 (by two out of the three authors), it is stated that $M(z,a)$ is analytic for $-6.71 \mu < a < 2 \mu /3 $. Does this mean it is analytic inside the unit disk in $z$ for all of these values of $a$ as well? What if we consider complex $a$, how much is known about analyticity (both in $a$ and $z$) in this case? The latter question would be important if one wishes to consider univalence for complex $a$.

  5. Below eq. (3.12) it is stated that "the physical domain for $a$ and $s_1$ are given by $[-2 \mu/9, 2 \mu/3)$ and $ [2 \mu/3, \infty) $" but that would correspond to $s \in [\mu, \infty) $ and $t \in [0,\mu]$ which is not really the physical range. As mentioned in point 1, the physical domain for $a$ would be $(-\infty, -2\mu/9)$ for the stated range of $s_1$.

In terms of more open questions - I have one: i. Szego theorem says that $r_1 > \frac{1}{4}$. Are there any theorems or anything known about $r_2, r_3$ etc? One would expect (as mentioned in the paper) that these would get closer to 1 as we gain more information about the function via its Taylor coefficients. The very nice plot (7b) seems to indicate the same as well.

In addition I found some typos/minor points: a. "brunch cuts" -> branch cuts on page 9, above equation (3.1)

b. "dispersion relations" -> dispersion relation also on page 9, above equation (3.3)

c. In the paragraph below eq (3.4) the discussion begins with generic $\mu$ but later when talking about the forward limit, $\mu = 4$ with the mass $m$ set to 1 is assumed without it being stated.

d. In formula (3.15), $\alpha$ on the RHS is not defined. I suppose it is the same as in https://arxiv.org/abs/2012.04877, where it is defined to be $\alpha = \frac{d-3}{2}$.

  • validity: high
  • significance: high
  • originality: high
  • clarity: high
  • formatting: excellent
  • grammar: perfect

Author:  Parthiv Haldar  on 2021-06-04  [id 1489]

(in reply to Report 4 on 2021-05-29)

  1. Regarding point 4, $\mathcal{M}(z,a)$ is analytic inside the unit disk in z for all of these values of -6.71μ<a<2μ/3. So far, the analytic properties in complex a have never been studied.

Referred to the open-ended question 1. The generalizations of Szego's theorem for $r_2, r_3$ etc are not known in general. However, there are some estimates for these in very few cases of sub-classes of the class $\mathcal{S}$.

---

## Editorial Decision

resubmitted